# Using adversarial networks to extend brain computer interface decoding accuracy over time

Xuan Ma[1†], Fabio Rizzoglio[1†], Kevin L Bodkin[1], Eric Perreault[2,3,4], Lee E Miller[1,2,3,4], Ann Kennedy[1]*

[1]Department of Neuroscience, Northwestern University, Chicago, United States; [2]Department of Biomedical Engineering, Northwestern University, Evanston, United States; [3]Department of Physical Medicine and Rehabilitation, Northwestern University, Chicago, United States; [4]Shirley Ryan AbilityLab, Chicago, United States

*For correspondence:
ann.kennedy@northwestern.edu

[†]These authors contributed equally to this work

Competing interest: The authors declare that no competing interests exist.

**Abstract** Existing intracortical brain computer interfaces (iBCIs) transform neural activity into control signals capable of restoring movement to persons with paralysis. However, the accuracy of the 'decoder' at the heart of the iBCI typically degrades over time due to turnover of recorded neurons. To compensate, decoders can be recalibrated, but this requires the user to spend extra time and effort to provide the necessary data, then learn the new dynamics. As the recorded neurons change, one can think of the underlying movement intent signal being expressed in changing coordinates. If a mapping can be computed between the different coordinate systems, it may be possible to stabilize the original decoder's mapping from brain to behavior without recalibration. We previously proposed a method based on Generalized Adversarial Networks (GANs), called 'Adversarial Domain Adaptation Network' (ADAN), which aligns the distributions of latent signals within underlying low-dimensional neural manifolds. However, we tested ADAN on only a very limited dataset. Here we propose a method based on Cycle-Consistent Adversarial Networks (Cycle-GAN), which aligns the distributions of the full-dimensional neural recordings. We tested both Cycle-GAN and ADAN on data from multiple monkeys and behaviors and compared them to a third, quite different method based on Procrustes alignment of axes provided by Factor Analysis. All three methods are unsupervised and require little data, making them practical in real life. Overall, Cycle-GAN had the best performance and was easier to train and more robust than ADAN, making it ideal for stabilizing iBCI systems over time.

## Editor's evaluation

This paper reports a new way to deal with the drift of neural signals and representations over time in a BCI. Given the context of the rapidly advancing field, the reviewers assessed the findings to be useful and potentially valuable. With the code provided for other investigators to use, the strength of evidence was convincing.

## Introduction

Intracortical brain-computer interfaces (iBCIs) aim to restore motor function in people with paralysis by transforming neural activity recorded from motor areas of the brain into an estimate of the user's movement intent. This transformation is accomplished using a neural 'decoder', an algorithm that translates the moment-to-moment activity of a population of neurons into a signal used to control intended movements. There has been substantial improvement in our ability to record and decode

from large populations of neurons in the past decade, which allows more information to be extracted from the brain and conveyed to the external effectors of the iBCI. However, the long-term stability of iBCIs is still far from satisfactory due in part to the instabilities in neural recordings. The relative micromotion between the electrode tip and the brain tissue (*Sussillo et al., 2016b*), the changes of regional extracellular environment (*Perge et al., 2013*), or even the active and inactive state shifts of neurons (*Volgushev et al., 2006*) could contribute to such instabilities, resulting in the turnover of signals picked by the chronically implanted electrodes on a time scale of days or even a few hours (*Downey et al., 2018*). Given these changes, a decoder could produce inaccurate predictions of the user's intent leading to the degraded iBCI performance.

To counteract these effects, a neural decoder might be recalibrated with newly acquired data. A disadvantage of this strategy is that during recalibration, normal use would be interrupted. Furthermore, the recalibration process likely means the user would need to learn the dynamics of the new decoder, imposing additional time and cognitive burden. For persons with paralysis to live more independently, an ideal iBCI would accommodate the gradual drift in neural recordings without supervision, thereby minimizing the need to periodically learn new decoders. For the performance of the initial 'day-0' decoder to be maintained, an additional component, an "input stabilizer", would need to be added to transform the neural recordings made on a later day ('day-k') such that they take on the statistics of the day-0 recordings.

Recently there has been a great deal of interest in the concept of a low-dimensional neural manifold embedded within the neural space that is defined by the full set of recorded neurons, and the 'latent signals' that can be computed in it (*Gallego et al., 2017*). A previous paper from our group demonstrated that by aligning the day-k and day-0 latent signals using canonical correlation analysis (CCA), the performance of a fixed day-0 decoder could be maintained over months and even years, despite turnover of the neural recordings.

Unfortunately, CCA has a couple significant limitations. For one, it is a linear process, not able to account for the nonlinear mappings that have been demonstrated between high-dimensional neural recordings and their low-dimensional manifolds (*Altan et al., 2021*; *Naufel et al., 2019*). Also, its use in a real-life scenario would be cumbersome. This application of CCA can be thought of as rotating two sets of neural signals 'spatially' to achieve optimal overlap (and thus temporal correlation). To do so requires cropping or resampling the single-trial data of behaviors on day-0 and day-k such that the paired trials correspond to the same behavior and contain the same number of timepoints, start condition, and end condition. Without trial-alignment, no amount of spatial rotation will achieve a correlation between the neural signals. However, motor behaviors in daily life are typically not well structured, with well-defined onsets and offsets, making trial alignment difficult, if not impossible. Where this method has been used successfully, it has been with highly stereotypic behaviors with distinct trial structure.

Another recently published linear method for decoder stabilization uses a Procrustes-based (*Gower and Dijksterhuis, 2004*) alignment on low-dimensional manifolds obtained from the neural activity using Factor Analysis (*Degenhart et al., 2020*). This approach, which we will refer to as 'Procrustes Alignment of Factors' (PAF), successfully stabilized online iBCI cursor control with a fixed decoder. Trial alignment is not needed for PAF, as it aligns the coordinate axes for the manifolds directly. However, it does require a subset of the coordinate axes in which the manifold is embedded (the neural recording channels) to be unchanged between days 0 and k. Furthermore, the use of a Procrustes-based transformation means that this strategy cannot correct for nonlinear changes in the neural manifold across days.

In another approach to decoder stabilization, we view changes in neural recordings as arbitrary shifts in the distribution of population firing rates. From this perspective, the reason for poor cross-day performance of decoders is clear: a decoder that is trained only on observations from a given distribution (e.g. those of 'day-0') won't perform well on data from other distributions (i.e. 'day-k'). A machine learning approach termed 'domain adaptation' has been used to cope with such distribution mismatches by learning a transformation that minimizes the difference between the transformed distributions; this permits a model trained on one distribution to generalize to another (*Farahani et al., 2021*; *Pan et al., 2011*). For example, if we have a classifier trained to distinguish photos of objects, domain adaptation could be used to transform drawings of those objects into 'photo-like' equivalents, so that the existing photo-based classifier could be used to distinguish the drawn objects.

Domain adaptation can be implemented with Generative Adversarial Networks (GANs; *Goodfellow et al., 2014*). GANs use two networks – a generator trained to transform a source distribution into a target distribution, and a discriminator trained to do the opposite: determine whether a given distribution is real or synthesized by the generator. The adversarial nature of the generator and discriminator enables the model to be trained in an unsupervised manner (*Ganin and Lempitsky, 2015*; *Tzeng et al., 2017*). GAN-based domain adaptation has been applied to computer vision problems, like adapting a classifier trained to recognize the digits of one style for use in recognizing those of another style (*Tzeng et al., 2017*), or translating images in the style of one domain to another (e.g. colorizing black-and-white photos, *Isola et al., 2017*).

We recently developed an approach we named Adversarial Domain Adaptation Network (ADAN; *Farshchian et al., 2018*), that used a GAN to perform domain adaptation to enable a fixed day-0 iBCI decoder to work accurately on input signals recorded on day-k. ADAN finds low-dimensional manifolds using a nonlinear autoencoder, and aligns the empirical distribution of the day-k recordings (the source domain) to those of day-0 (the target domain) by aligning the distributions of residuals (as in *Zhao et al., 2016*) between neural firing rates and their nonlinear autoencoder reconstructions (that is, the portion of neurons' activity not predicted from the manifold). Note that, compared to PAF, ADAN performs the alignment in the high-dimensional space of reconstructed firing rates, but requires the computation of a low-dimensional manifold to do so. In the earlier study we found that ADAN outperforms both CCA and an alignment process that minimized the KL divergence between the distributions of the day-k and day-0 latent spaces (Kullback-Leibler Divergence Minimization, KLDM; *Farshchian et al., 2018*). However, ADAN was only tested on data from a single monkey and a single task, for just 2 weeks. Our subsequent exploration into applying ADAN to other datasets suggests that, while it can work in other settings, its performance is quite sensitive to model hyperparameter settings. This is consistent with previous reports that GANs can be highly dependent on choice of architecture and a variety of hyperparameter settings (*Farnia and Ozdaglar, 2020*). We therefore sought alternative GAN-based approaches that might offer more robust performance.

Recently, *Zhu et al., 2017* developed a novel GAN architecture named Cycle-Consistent Adversarial Networks (Cycle-GAN) in the context of image domain adaptation. Cycle-GAN introduced a mechanism termed 'cycle-consistency', which helps to regularize model performance. Specifically, Cycle-GAN implements both forward and inverse mappings between a pair of domains: the forward mapping translates data in the source domain to the target domain, while the inverse mapping brings the translated data back to the source domain. This regularization mechanism forces the learned transformation between the source and the target distributions to be a bijection, thereby reducing the search space of possible transformations (*Almahairi et al., 2018*; *Zhu et al., 2017*).

In addition to its promise of greater robustness, Cycle-GAN is to our knowledge unique among neural alignment methods in that it does not rely on projection of neural population activity to a low-dimensional manifold: rather, it aligns the full-dimensional distributions of the day-0 and day-k recordings directly. Other alignment methods that we have explored (CCA, PAF, KLDM, and ADAN) all work with low-dimensional latent signals. Aligning on full-dimensional data leads to the advantage that the (small) information loss caused by dimensionality reduction can be avoided. Furthermore, as most existing iBCI decoders are computed directly from the full-dimensional neural recordings, no extra transformation of neural recordings is required between alignment and decoding.

In this study, we compare Cycle-GAN, ADAN, and PAF using datasets from several monkeys, spanning a broad variety of motor behaviors, and spanning several months. We chose not to test CCA, as it requires trial alignment of the data, and it (as well as KLDM) was outperformed by ADAN in our earlier study (*Farshchian et al., 2018*). We found that both GAN-based methods outperformed PAF. We also demonstrated that the addition of cycle-consistency improved the alignment and made training much less dependent on hyperparameters.

## Results

### Performance of a well-calibrated iBCI decoder declines over time

We trained six monkeys to perform five tasks: power and key grasping, center-out target reaching using isometric wrist torque, and center-out and random-target reaching movements (*Figure 2—figure supplement 1*). After training, each monkey was implanted with a 96-channel microelectrode array in

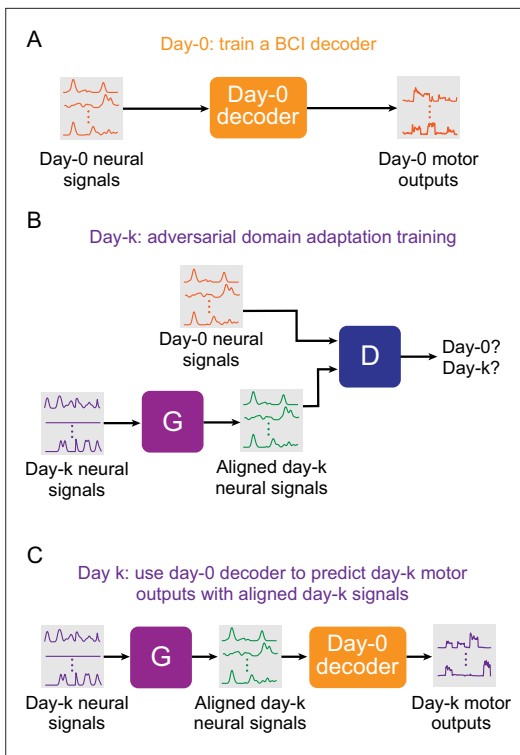

**Figure 1.** Setup for stabilizing an intracortical brain computer interface (iBCI) with adversarial domain adaptation. (**A**) Initial iBCI decoder training on day-0. The decoder is computed to predict the motor outputs from neural signals, using either the full-dimensional neural recordings or the low-dimensional latent signals obtained through dimensionality reduction. This decoder will remain fixed over time after training. (**B**) A general framework for adversarial domain adaptation training on a subsequent day-k. The 'Generator' (G) is a feedforward neural network that takes day-k neural signals as the inputs and aims to transform them into a form similar to day-0 signals; we also refer to G as the 'aligner'. The 'Discriminator' (D) is another feedforward neural network that takes both the outputs of G (aligned day-k neural signals) and day-0 neural signals as the inputs and aims to discriminate between them. (**C**) A trained aligner and the fixed day-0 decoder are used for iBCI decoding on day-k. The aligned signals generated by G are fed to the day-0 decoder to produce the predicted motor outputs.

either the hand or arm area of M1. Four animals (monkeys J, S, G, P) were also implanted with intramuscular leads in forearm and hand muscles contralateral to the cortical implant; these were used to record electromyograms (EMGs). We recorded multi-unit activity on each M1 electrode together with motor output (EMGs and/or hand trajectories) for many sessions across multiple days. All recording sessions for a specific task and an individual monkey were taken together to form a dataset. We collected a total of seven datasets, and the recording sessions in each of them spanned from ~30 to~100 days (See Materials and methods; *Figure 2—source data 1*).

As in previous studies (*Gallego et al., 2020*; *Sussillo et al., 2016b*), we found substantial instability in the M1 neurons we recorded over time, even though the motor outputs and task performance were generally stable (*Figure 2— figure supplements 2 and 3*). We first asked how this instability affected the performance of an iBCI decoder. We fit a Wiener filter decoder with data recorded on a reference day (designated 'day-0'; *Figure 1A*). We then used this decoder to predict the motor outputs from M1 neural recordings on later days ('day-k') and computed the coefficient of determination ($R^2$) between the predictions and the actual data (see Materials and methods). *Figure 2* shows example predictions from each task. In all cases, both EMG (top row) and kinematic (bottom row) decoders could reconstruct movement trajectories with high accuracy on held-out trials from the day of training ('day-0'). However, the calibrated day-0 decoders consistently failed to predict EMGs or hand trajectories accurately on day-k. The degradation of the performance across time occurred for all behavioral tasks and monkeys, and could be substantial even a few days after decoder training (*Figure 2— figure supplement 4*).

## Adversarial networks mitigate the performance declines of day-0 decoders

We proposed to use generative adversarial network (GAN) based domain adaptation (*Figure 1B*) to address the problem described above. We tested two different architectures: Adversarial Domain Adaptation Network (ADAN) (*Farshchian et al., 2018*), and Cycle-Consistent Adversarial Networks (Cycle-GAN) (*Zhu et al., 2017*). As both ADAN and Cycle-GAN were trained to reduce the discrepancy between the neural recordings on day-0 and those on day-k by aligning their probability density functions (PDFs), we call them 'aligners'. Importantly, both ADAN and Cycle-GAN are static methods, trained only on instantaneous neural activity datapoints with no knowledge of dynamics. Both methods are causal and can be used in real time. We used the dataset with the longest recording timespan (monkey J, isometric wrist task, spanning 95 days) to determine appropriate choices of the hyperparameters for neural network training, which are presented in detail in a later section. We used

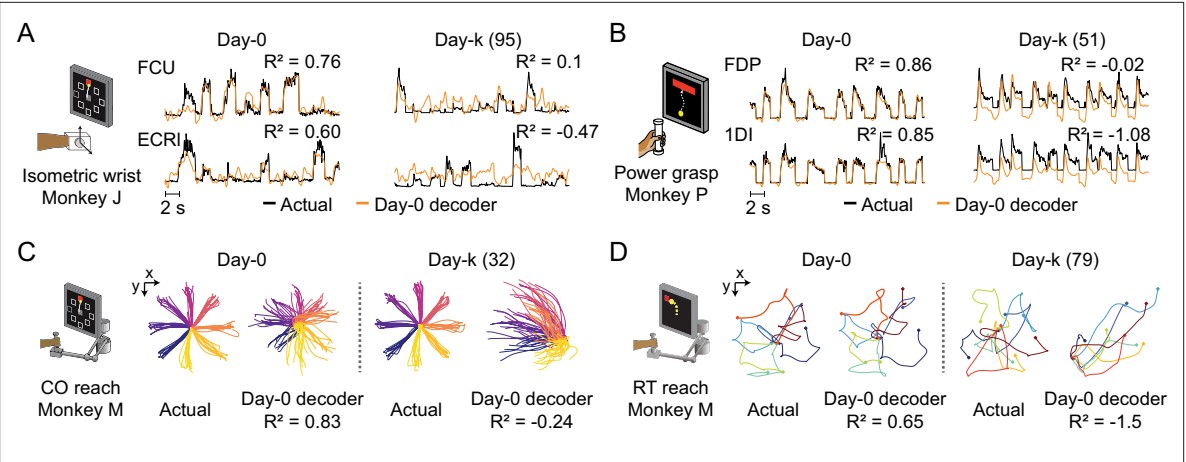

**Figure 2.** The performance of well-calibrated decoders declines over time. (**A**) Actual EMGs (black) and predicted EMGs (orange) using the day-0 decoder for flexor carpi ulnaris (FCU) and extensor carpi radialis longus (ECRl) during the isometric wrist task. (**B**) Actual and predicted EMGs using the day-0 decoder for flexor digitorum profundus (FDP) and first dorsal interosseous (1DI) during the power grasp task. (**C**) Actual hand trajectories and predictions using the day-0 decoder during the center-out (CO) reach task. Colors represent different reaching directions. (**D**) Actual and predicted hand trajectories using the day-0 decoder during the random-target (RT) reach task. Colors represent different reaching directions.

The online version of this article includes the following source data and figure supplement(s) for figure 2:

**Source data 1.** Table summarizing the datasets analyzed in this paper, including cortical implant site and date, number of recording sessions, number of days between recording start and end, recording days relative to time of array implantation, and motor outputs (EMG or hand velocities) recorded.

**Figure supplement 1.** Behavior tasks.

**Figure supplement 2.** Unstable neural recordings underlying stable motor outputs.

**Figure supplement 3.** Evaluation of the stability of M1 neural signals and motor outputs over time for monkeys / tasks (besides monkey J).

**Figure supplement 4.** The accuracy of a well-calibrated iBCI decoder degrades over time for different behavioral tasks.

the resulting hyperparameter values for the tests of all other monkeys and tasks. For comparison, we also used all datasets to test another type of 'aligner' that aimed to align the low-dimensional neural manifolds between day-0 and day-k (*Degenhart et al., 2020*), which we termed 'Procrustes Alignment of Factors' (PAF).

The tests were conducted with the procedures presented by *Figure 1*. First, we picked a given day as day-0, and used the data recorded on that day to fit a Wiener filter as the 'day-0 decoder' (*Figure 1A*). Then, we trained the three types of aligners (ADAN, Cycle-GAN, and PAF) to align the neural recordings on a different day (day-k) to those on day-0 (*Figure 1B*). Each day in a dataset other than the designated day-0 was treated as a day-k, whether it occurred before or after day-0. Finally, we processed the neural recordings on day-k with the trained aligners, fed the aligned signals to the fixed day-0 decoder, and evaluated the accuracy of the predictions this decoder could obtain (*Figure 1C*). For each of the seven datasets being tested, we repeated these three procedures for multiple instantiations using different day-0s (see *Figure 2—source data 1*). To characterize the performance of the day-0 decoder after alignment, we represent the decoder accuracy as the 'performance drop' with respect to a daily recalibrated decoder ($R^2_{aligned} - R^2_{same-day}$). If an aligner works perfectly, we expect the performance drop of day-0 decoders to be close to 0, which means the decoder achieves accuracy equal to a within-day decoder after the alignment.

Unlike ADAN and PAF, Cycle-GAN alignment does not require computation of a latent representation from neural recordings. As a result, Cycle-GAN is naturally suited to a decoder trained on the full-dimensional neural firing rate signals. It is theoretically possible to use a full-dimensional decoder with ADAN and PAF as well, by training on firing rates reconstructed from the latent spaces of the ADAN autoencoder and PAF factors respectively. However, we found that the performance of these full-dimensional decoders was inferior to that of a decoder trained on the inferred latent signals (*Figure 3—figure supplement 1*). For completeness, we also tested a decoder trained on Cycle-GAN-generated firing rates projected into a low-dimensional manifold obtained using Factor

Analysis; as expected, its performance was slightly worse than that of a full-dimensional decoder, but still better than ADAN and PAF with a low-dimensional decoder (*Figure 3—figure supplement 1*).

In light of the analysis above, we here compare the better-performing of the two potential decoder input formats for each alignment method: full-dimensional for Cycle-GAN, and low-dimensional for ADAN and PAF (*Figure 3*, see Materials and methods for details). Aside from this difference of input dimensionality, the architecture of the day-0 decoder (a Wiener filter) was the same for all aligners. The within-day accuracy of the day-0 decoders of the three aligners was modestly but significantly different across tasks (*Figure 3A*): ADAN: $R^2$=0.73 ± 0.009 (mean ± s.e.); Cycle-GAN: $R^2$=0.72 ± 0.009; PAF: $R^2$=0.71 ± 0.009 (p=0.008, linear mixed-effect model with the type of aligner as fixed and the type of task as random factor, n=204 samples, where each sample is one aligner/task/day-0/day-k combination).

To test for a significant performance difference between aligners, we fit a linear mixed-effect model with type of aligner and days as fixed factors and type of task as random factor for a quantitative evaluation of the performance of the three aligners (n=2361 samples). The performance drop of the day-0 decoder on data collected on the day immediately following day-0 (i.e. day-1) after alignment was significantly different across the aligners (Cycle-GAN: –0.02±0.004 (mean ± s.e.); ADAN: –0.06±0.005; PAF: –0.11±0.005; p~0). Cycle-GAN significantly outperformed both ADAN (p~0) and PAF (p~0). ADAN also significantly outperformed PAF (p~0).

The performance degradation of day-0 decoders for periods greater than one day (*Figure 2—figure supplement 4*) was also mitigated by all three alignment methods, although to different extents. Nonetheless, there remained a significant and increasing performance drop over time (*Figure 3A and B*). We found a significant interaction between time and alignment method (p=0.026), indicating that there was a difference between methods in performance drop over time, and a post-hoc comparison showed that Cycle-GAN had the least overall performance degradation, significantly better than PAF, and better, but not significantly so, than ADAN (p=0.008 vs PAF; p=0.328 vs ADAN). ADAN was better, but not significantly, than PAF (p=0.091). Taken together, this analysis shows that Cycle-GAN moderately outperforms both ADAN and PAF (see also *Figure 3C*; *Figure 3—figure supplement 2B, C*), and furthermore that the two nonlinear alignment methods tend to be more stable over time than PAF (see also *Figure 3C*; *Figure 3—figure supplement 2A, B*).

While CCA-style trial alignment is not required by Cycle-GAN, ADAN, or PAF, we did preprocess the data to exclude behaviors not related to the investigated task (inter-trial data) and used data only from the beginning to the end of each trial (see Materials and methods). Among other advantages, this helped to unify behavior across monkeys and behavioral tasks. However, in a true iBCI setting, the user has uninterrupted control, so it would be ideal to train the aligner on that data, without the need to classify and exclude portions of a recording session that are not task-related. Therefore, we also tested aligners on the continuous neural recordings on the isometric wrist task data of monkey J (*Figure 3—figure supplement 3*). Under this condition, Cycle-GAN was clearly superior to ADAN and PAF. We fit a linear mixed-effect model with type of aligner and days as fixed factors (n=531 samples) and found that the accuracy of the day-0 decoder on day-1 after alignment was significantly different across the aligners (Cycle-GAN: –0.05±0.015 (mean ±s.e.); ADAN: –0.14±0.023; PAF: –0.18±0.019; p~0). Cycle-GAN significantly outperformed both ADAN (p~0) and PAF (p~0), while ADAN outperformed PAF, but not significantly (p=0.134). On the other hand, we did not find a significant interaction between time and alignment method (p=0.56), indicating that the performance degradation over time was mitigated in a similar way by all three methods.

## Cycle-GAN is robust to hyperparameter settings

While they can be powerful, GANs can present a training challenge: choosing suitable hyperparameters is important, for example, to balance the learning process and prevent either of the two networks (the generator or discriminator) from dominating the loss function. High sensitivity of model performance to hyperparameter values would pose a potential barrier to the adoption of either ADAN or Cycle-GAN as a tool for cross-day alignment. As in *Ghosh et al., 2020*, we assessed sensitivity to hyperparameters by testing the impact of batch size and learning rates on alignment performance. Because these hyperparameter sweeps are very computationally expensive, we evaluated them using only the single dataset with the greatest span of time.

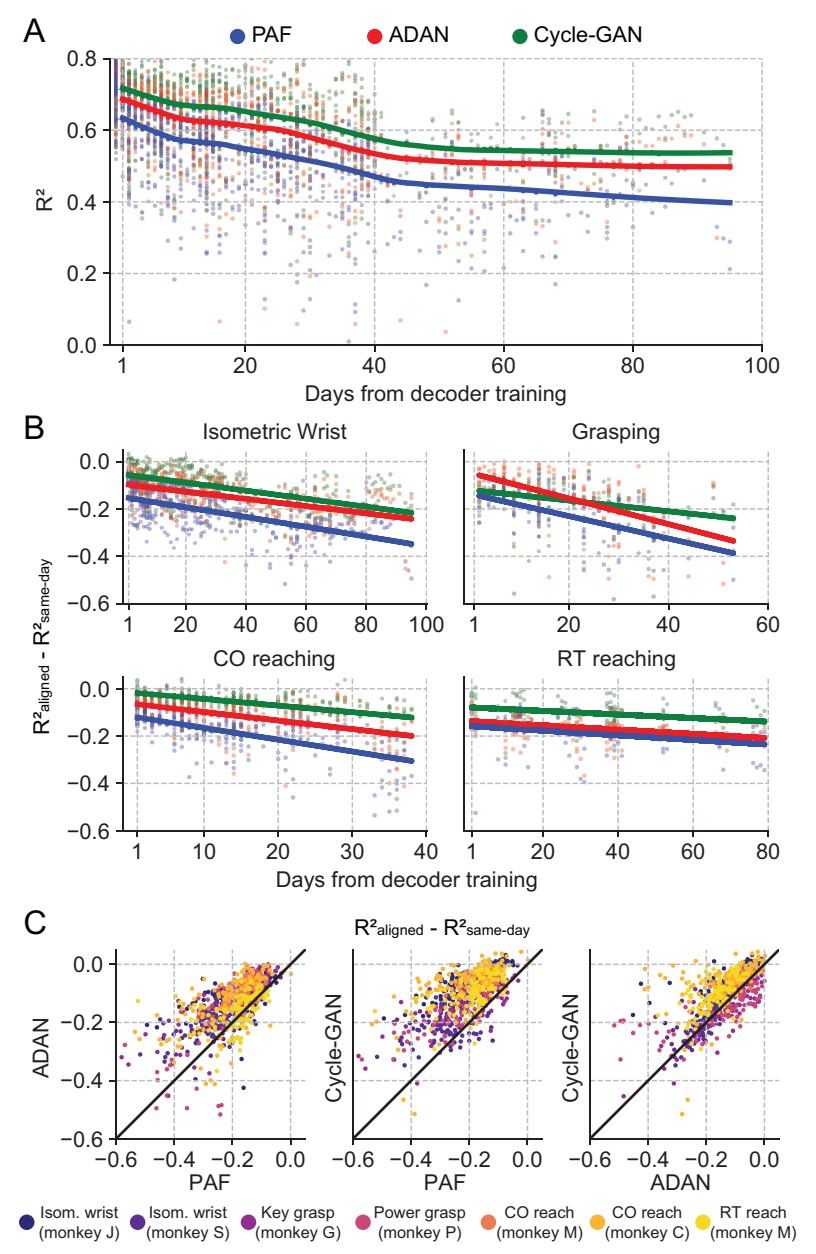

**Figure 3.** The proposed GANs-based domain adaptation methods outperform Procrustes Alignment of Factors in diverse experimental settings. (**A**) Prediction accuracy over time using the fixed decoder trained on day-0 data is shown for all experimental conditions (single dots: $R^2$ as a function of days after decoder training, lines: locally weighted scatterplot smoothing fits). We compared the performance of the day-0 decoder after domain adaptation alignment with Cycle-GAN (green), ADAN (red) and PAF (blue). (**B**) We computed the prediction performance drop with respect to a daily-retrained decoder (single dots: $R^2$ drop ($R^2_{aligned} - R^2_{same-day}$) for days after decoder training, lines: linear fits). Cycle-GAN and ADAN both outperformed PAF, with Cycle-GAN degrading most slowly for all the experimental conditions. (**C**) We compared the performance of each pair of aligners by plotting the prediction performance drop of one aligner versus that of another. Each dot represents the $R^2$ drop after decoder training relative to the within-day decoding. Marker colors indicate the task. Both proposed domain adaptation techniques outperformed PAF (left and center panels), with Cycle-GAN providing the best domain adaptation for most experimental conditions (right panel).

The online version of this article includes the following figure supplement(s) for figure 3:

**Figure supplement 1.** Cycle-GAN outperforms ADAN and Procrustes Alignment of Factors (PAF) with both full-dimensional and low-dimensional day-0 decoder.

*Figure 3 continued on next page*

*Figure 3 continued*

**Figure supplement 2.** Cycle-GAN and ADAN consistently outperform Procrustes Alignment of Factors (PAF) for all experimental conditions.

**Figure supplement 3.** Cycle-GAN outperforms ADAN and Procrustes Alignment of Factors (PAF) when aligning continuous neural recordings.

We trained both ADAN and Cycle-GAN aligners on day-k data relative to four selected day-0 reference days. We kept the learning rates for the generator ($LR_G$) and the discriminator ($LR_D$) fixed (for ADAN, $LR_G = 0.0001$, $LR_D/LR_G = 0.5$; for Cycle-GAN, $LR_G = 0.0001$, $LR_D/LR_G = 10$). As in the previous section, we evaluated the drops in aligned day-0 decoder accuracy. We found that ADAN maintained good performance when batch size was small, but that performance started to drop significantly for larger batch sizes (64: $-0.13\pm0.0096$ (mean ± s.e.); 256: $-0.17\pm0.013$; p~0, Wilcoxon's signed rank test, n=76; *Figure 4A*). In contrast, Cycle-GAN based aligners performed consistently at all tested batch sizes. These results suggest that ADAN may need a small batch size, while Cycle-GAN-based aligners have no strong requirement.

Neural network training time is inversely proportional to batch size - therefore given two batch size options that give comparable model performance, the larger of the two will yield faster training. We found that Cycle-GAN was slower than ADAN for smaller batch sizes, although neither method

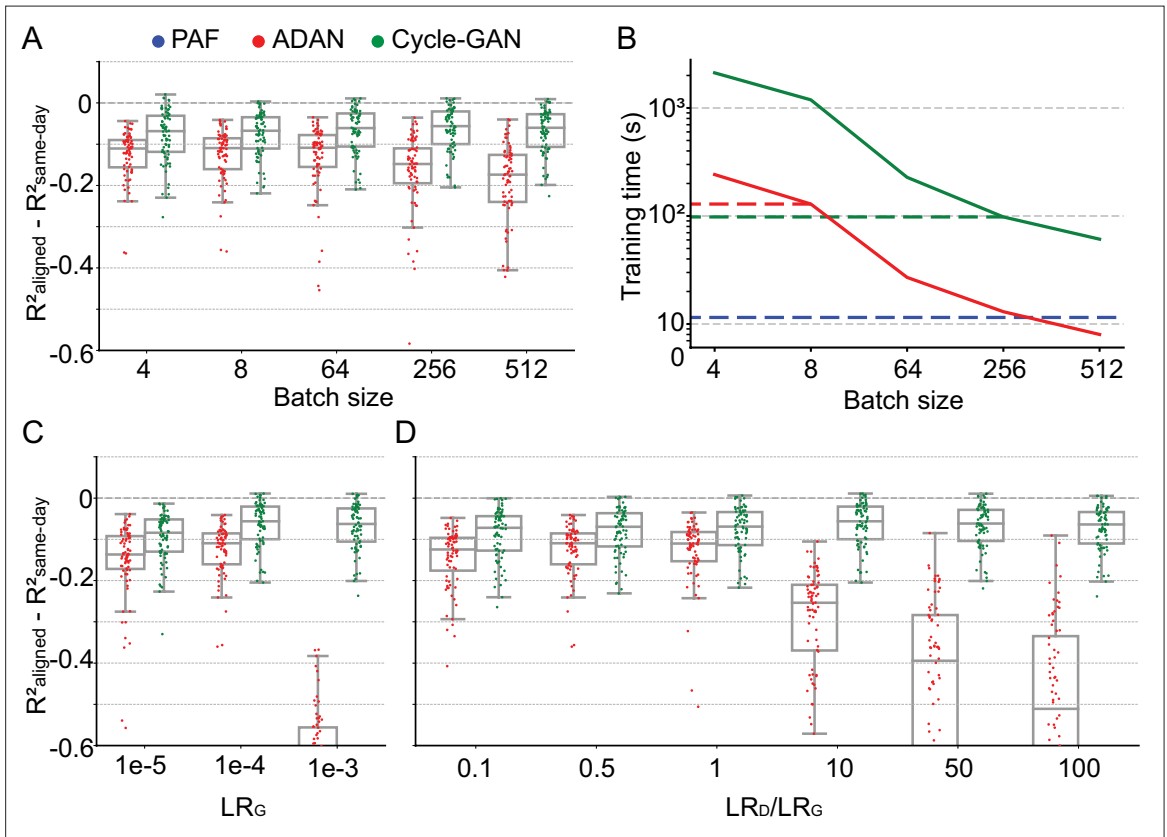

**Figure 4.** Cycle-GAN is more robust to hyperparameter tuning than ADAN. Effect of different batch sizes during training of Cycle-GAN (green) and ADAN (red) with mini-batch gradient descent on (**A**) the day-k performance of 4 selected day-0 decoders and (**B**) the execution time of 200 training epochs. The much faster execution time of PAF (blue) is also shown for reference. Compared to ADAN, Cycle-GAN did not require a small batch size, resulting in faster training (Cycle-GAN: 98 s with batch size 256; ADAN: 129 s with batch size 8; FA aligner: 11.5 s). Effect of training each domain adaptation method with different generator (**C**) and discriminator (**D**) learning rate. The generator and the discriminator learning rate were denoted as $LR_G$ and $LR_D$, respectively. For $LR_D$ testing, we kept $LR_G$ fixed ($LR_G = 1e-4$ for both ADAN and Cycle-GAN), and changed the ratio between $LR_D$ and $LR_G$ ($LR_D/LR_G$). ADAN-based aligners did not perform well for large $LR_G$ or $LR_D/LR_G$ values, while Cycle-GAN-based aligners remained stable for all the testing conditions. In (**A**), (**C**) and (**D**) single dots show the prediction performance drop on each day-k relative to the 4 selected day-0s with respect to the $R^2$ of a daily-retrained decoder ($R^2_{aligned} - R^2_{same-day}$). Boxplots show 25th, 50th and 75th percentiles of the $R^2$ drop with the whiskers extending to the entire data spread, not including outliers.

required more than a few minutes when operating within their optimal batch size range (*Figure 4B*). Thus, we set the ADAN batch size for subsequent analyses to 8 and for Cycle-GAN to 256. Although we could have increased the batch size for ADAN, we decided instead to use a conservative value further from its region of decreased performance at the expense of slower training. For reference, we also computed the execution time of PAF, which was much faster than both ADAN and Cycle-GAN (*Figure 4B*, dashed blue line) as it has a closed form solution (*Schönemann, 1966*). We also note that the inference time (i.e. the time it takes to transform data once the aligner is trained) for both Cycle-GAN and ADAN is well under 1 ms per 50 ms sample of neural firing rates– this is because the forward map in both models consists simply of a fully connected network with only two hidden layers.

We next examined the effect of learning rates for each aligner. We first tested different values for the $LR_G$, while fixing the ratio between $LR_D$ and $LR_G$ (for ADAN, $LR_D/LR_G = 0.5$; for Cycle-GAN, $LR_D/LR_G = 10$). As shown in *Figure 4C*, ADAN achieved good performance when $LR_G$ was set to 1e-5 and 1e-4 but did not work well if $LR_G$ was set to 1e-3. Cycle-GAN maintained stable performance when $LR_G$ was set to 1e-3 and 1e-4, and had a significant performance drop when $LR_G$ was 1e-5 (1e-4: $-0.064\pm0.0062$ (mean $\pm$ s.e.); 1e-5: $-0.095\pm0.0068$; p~0, Wilcoxon's signed rank test, n=76), but still significantly better than ADAN with the same $LR_G$ (Cycle-GAN: $-0.095\pm0.0068$ (mean $\pm$ s.e.); ADAN: $-0.15\pm0.011$; p~0, Wilcoxon's signed rank test, n=76). We then tested different ratios between $LR_D$ and $LR_G$ with $LR_G$ fixed ($LR_G$ = 1e-4 for both types of aligners). As *Figure 4D* shows, ADAN could only be trained well when $LR_D$ was equal to or smaller than $LR_G$. On the other hand, the performance of a Cycle-GAN based aligner remained stable for all tested $LR_D/LR_G$ values.

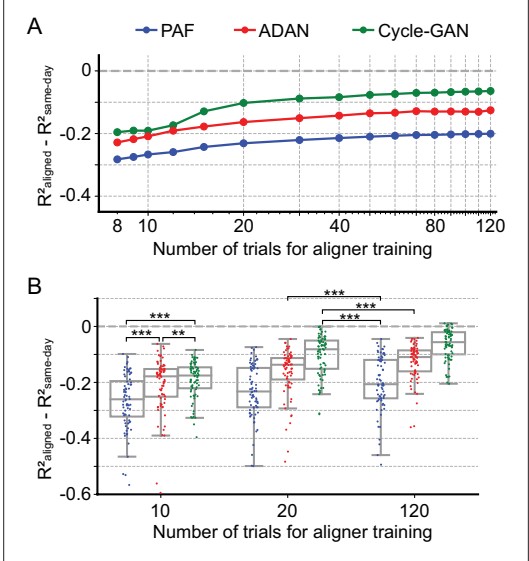

**Figure 5.** Cycle-GAN and ADAN need only a limited amount of data for training. (**A**) Effect of the number of trials used for training Cycle-GAN (green), ADAN (red) and PAF (blue) on the day-k decoding accuracy using 4 selected day-0 fixed decoders. All the aligners needed 20–40 trials to achieve a satisfactory performance, before reaching a plateau. The average prediction performance drop with respect to a daily-retrained decoder ($R^2_{aligned}$ - $R^2_{same-day}$) on all day-ks is shown for each tested value of training trials (x-axis is in log scale). When using 10 trials, both Cycle-GAN and ADAN significantly outperformed PAF (**B**, left boxplots). Moreover, both Cycle-GAN-based and ADAN aligners trained with 20 trials had significantly better performance than the PAF trained on all 120 trials (**B**, center and right boxplots). Single dots show the prediction performance drop on each day-k to the 4 selected day-0s with respect to a daily-retrained decoder. Boxplots show 25th, 50th and 75th percentiles of the $R^2$ drop with the whiskers extending to the entire data spread, not including outliers. Asterisks indicate significance levels: *p<0.05, **p<0.01, ***p<0.001.

## GAN-based methods require very little training data for alignment

Aligners in practical iBCI applications must be fast to train and perhaps more importantly, require little training data. Here we investigated the aligner performance with limited training data. We trained ADAN, Cycle-GAN, and PAF to align the data on each day-k to four selected day-0s using randomly selected subsets of the full 120-trial training set from Monkey J. We then decoded EMGs from the aligned M1 signals on a fixed 40-trial held-out testing set using the day-0 decoder. As *Figure 5A* shows, all three aligners improved the performance of day-0 decoders with 20 or fewer training trials. Performance increased as more training trials were included but started to plateau near 40 trials. When using only 10 trials, both ADAN and Cycle-GAN significantly outperformed PAF (Cycle-GAN: $-0.19\pm0.0076$ (mean $\pm$ s.e.); ADAN: $-0.21\pm0.011$; PAF: $-0.26\pm0.011$; p~0, Wilcoxon's signed rank test, n=76), with Cycle-GAN significantly outperforming ADAN (p=0.003, Wilcoxon's signed rank test, n=76). It is also worth noting that ADAN and Cycle-GAN trained with only 20 trials significantly

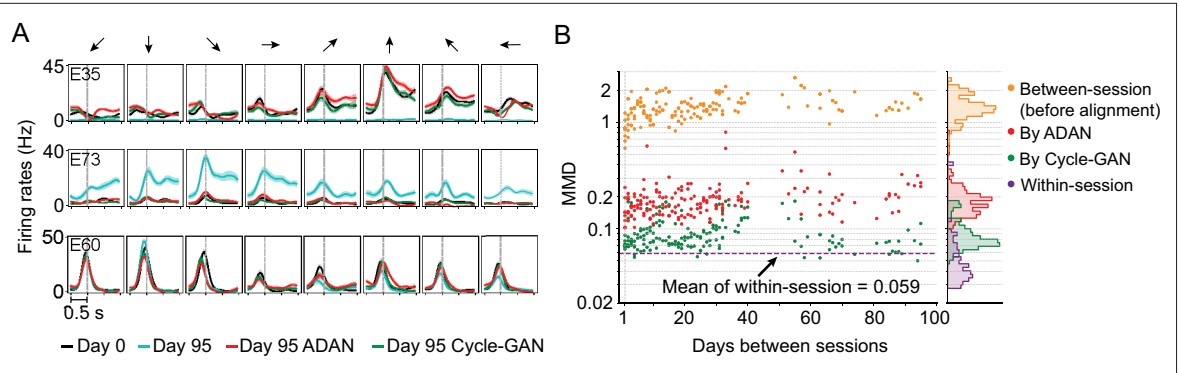

**Figure 6.** The changes of single-electrode and coordinated neural activity patterns after alignment. (**A**) The PETHs of the multiunit activity from three cortical electrodes (E35, E73, E60) before and after alignment. Each column corresponds to a target direction indicated by the arrows on the top. For each direction, mean (solid lines) and standard errors (shaded areas) are shown for 15 trials. The dashed vertical line in each subplot indicates the time of force onset. (**B**) Between-session MMDs for M1 signals before and after alignment, as well as the within-session MMDs. The main panel plots the between-session MMDs before (orange) and after alignment (red: by ADAN, green: by Cycle-GAN) for all pairs of sessions with different days apart, and the dashed purple line indicates the mean of the within-session MMD values. The side panel plots the histogram for each type of data. Note y-axis is in log scale.

outperformed PAF trained with the full training set of 120 trials (Cycle-GAN trained with 20 trials: $-0.10\pm0.0083$ (mean $\pm$s.e.); ADAN trained with 20 trials: $-0.16\pm0.0096$; PAF trained with 120 trials: $-0.20\pm0.011$; p~0, Wilcoxon's signed rank test, n=76) (*Figure 5B*).

## Recovery of single-electrode activity patterns through alignment

Both ADAN and Cycle-GAN generate reconstructed versions of the aligned day-k single neuron signals, agnostic to downstream use. However, our objective of decoder stabilization does not require that the full distribution of day-0 responses be recovered: we need only recover signals that are relevant to the decoding dimension. Decoder performance alone therefore does not provide a complete picture of the quality of neural alignment. To more thoroughly investigate the extent to which distribution alignment introduces biases or artifacts in predicted neural responses, we first compared aligner predictions of single-neuron with those of their recorded day-0 analogs.

Because PAF operates directly on the low-dimensional neural manifold, it can only generate single-neuron responses in the aligned representation by projecting back out from the manifold. We found that a stabilized day-0 decoder that uses these reconstructed firing rates from the latent space of the PAF factors performs poorly (*Figure 3—figure supplement 1C*). In contrast to PAF, Cycle-GAN and ADAN each generate synthetic firing rates for the full neural population (although ADAN still relies on a low-dimensional manifold as an intermediate step). Therefore, we restricted our analysis of single-neuron properties on the outputs of ADAN and Cycle-GAN.

Specifically, we asked how response properties of the day-k 'aligned neurons' differed from those of the neurons recorded on the same electrode on day-0. To do so, we examined the aligned neural representations generated by Cycle-GAN and ADAN, again using the 95-day isometric wrist task dataset of monkey J. We first compared the peri-event time histograms (PETHs) of firing rates before and after alignment, to determine how the aligners altered day-k neural activity at the level of single electrodes. The PETHs in *Figure 6A* show three examples of the ways in which single-electrode signals may differ across days, and the change produced by alignment. Electrode E35 is an example of neuron drop-out, in which the activity captured on day-0 was not observed on day-95. The PETHs of aligned day-95 data matched those of day-0 for all force directions, demonstrating that on day 95 both ADAN and Cycle-GAN aligners synthesized appropriate neural activity (*Figure 6A*). Second, E73 is an example of activity not present on day-0, but recorded on day-95. In this case, the day-95 activity was suppressed to match that on day-0. Finally, E60 is an example of consistent neural activity over the two days, which the aligners left unchanged.

We also examined the distributions of the recovered single-electrode activity by computing the Maximum Mean Discrepancy (MMD *Gretton et al., 2012a*, see Materials and methods) between all pairs of sessions (*Figure 6B*). Before alignment, the between-day MMDs were significantly larger

than the within-day MMDs (orange, between-day MMD: 1.42±0.029 (mean ± s.e.); purple, within-day MMD: 0.059±0.0054; p~0, Wilcoxon's rank sum test, n=171). After alignment, the between-day MMDs were substantially reduced by both Cycle-GAN and ADAN, becoming comparable to the within-day MMDs (ADAN: red, 0.19±0.0065 (mean ±s.e.); Cycle-GAN: green, 0.091±0.0024; within-day: purple, 0.059±0.0054). Cycle-GAN based aligners generally achieved a significantly lower between-day MMD than ADAN across the entire timespan (p~0, Wilcoxon's rank sum test, n=171).

## Recovery of neural manifolds from aligned representations

While Cycle-GAN works only with the full-dimensional neural recordings, ADAN, whose discriminator is essentially an autoencoder, computes a low-dimensional neural manifold from which it reconstructs the high-level signals it needs to align the high-level residuals. Consequently, we wanted to explore to what extent each method also altered the low-dimensional representations. We applied Principal Component Analysis (PCA) to the firing rates recorded for the 95-day isometric wrist task of monkey J on four selected day-0s and examined the trajectories of M1 neural activity within the neural subspaces defined by the principal components (PCs, see Materials and methods). We then projected the firing rates of the remaining day-k's onto the neural subspace defined by the corresponding day-0 PCs.

Generally, the day-k neural trajectories projected onto the top two day-0 PCs did not match those of day-0 (*Figure 7A*). However, after alignment (3rd and 4th columns), the day-k trajectories closely resemble those of day-0.

Finally, to directly quantify the similarity between the neural manifolds of day-0 and an aligned day-k, we calculated the principal angles (*Knyazev and Argentati, 2002*) between the neural subspaces for all sessions relative to the selected day-0 (see Materials and methods). To interpret the magnitude of the overlap between a given pair of days, we compared the observed angle with an upper bound provided by the principal angles across random subspaces that preserved the covariance of the day-0 and day-95 neural data, using the method described in *Elsayed et al., 2016*. We also found a 'within-day' bound by computing the angles between the day-0 neural recordings of even-numbered trials and odd-numbered trials (this was done to reduce the effect of any within-day drift). We found that alignment with either Cycle-GAN or ADAN made the neural manifolds of any day-k substantially more similar to those of day-0. In particular, after applying Cycle-GAN-based aligners, the population subspaces highly overlapped (*Figure 7B*).

## Discussion

We previously demonstrated the utility of a GAN-based method, ADAN, to 'align' M1 data across time, thereby allowing a fixed iBCI decoder to be used for weeks without re-calibration, despite a gradual change in the neurons recorded over the same period (*Farshchian et al., 2018*). However, we had tested ADAN on a very limited dataset. Because GANs are notoriously sensitive to hyperparameter settings (*Farnia and Ozdaglar, 2020*; *Ghosh et al., 2020*; *Kurach et al., 2018*), it was unclear how robust ADAN would be in practice. Another promising method, PAF, had been tested primarily in terms of two monkeys' online iBCI performance (*Degenhart et al., 2020*). We wished to compare both approaches directly, using a very diverse dataset including recordings from six monkeys and five tasks. We also compared a third approach based on a more recent GAN architecture, Cycle-GAN (*Zhu et al., 2017*). Cycle-GAN has the potential advantage over ADAN that it reduces the search space of aligners by encouraging the learned transformation to be a bijection, which might help stabilize its performance. Moreover, unlike ADAN and PAF, the Cycle-GAN architecture does not require computation of a low-dimensional manifold underlying the neural population activity, allowing its straightforward use with spike-rate based decoders.

Both ADAN and Cycle-GAN achieved higher performance than PAF, but each method had tradeoffs. Although ADAN needed less time to train than Cycle-GAN, PAF was much faster to train than both GAN methods. But while slower, Cycle-GAN was easier to train than ADAN, in the sense that it was less sensitive to hyperparameter values and therefore likely to be more effective 'out-of-the-box', and when working with different data binning and sampling rates. Importantly, Cycle-GAN also had clearly superior performance compared to both ADAN and PAF when tested with continuously recorded data (with no trial segmentation). Overall, our work suggests that GAN-based alignment, and Cycle-GAN in particular, is a promising method for improving the stability of an iBCI over time.

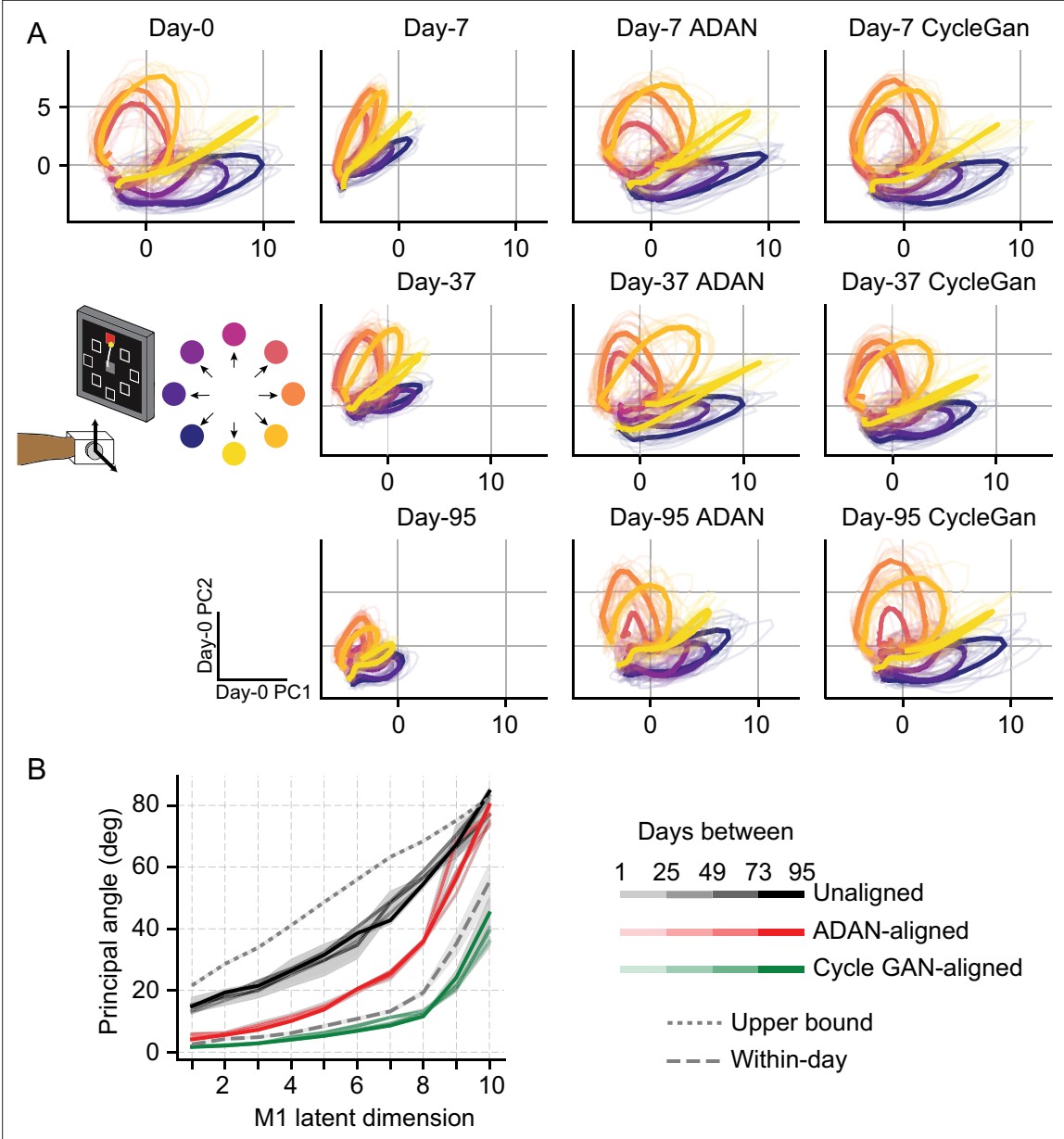

**Figure 7.** Neural manifold is stable over time after domain adaptation based neural alignment. (**A**) Representative latent trajectories when projecting unaligned / aligned neural activity onto the first two principal components (PCs) for the day-0 neural activity of monkey J during isometric wrist task. Top left corner: latent trajectories for day-0 firing rates, as the reference. 2nd column: latent trajectories for unaligned firing rates on day-7 (top row), day-37 (center row) and day-95 (bottom row). 3rd column and 4th column: latent trajectories for firing rates aligned by ADAN (3rd column) and Cycle-GAN (4th column) on day-7, day-37, and day-95. Data were averaged over the first 16 trials for each target location and aligned to movement onset for visualization purposes. (**B**) First ten principal angles between the neural manifolds of day-0 and a given day-k for unaligned (black), aligned by ADAN (red) and aligned by Cycle-GAN (green). Upper bound was found by computing principal angles between surrogate subspaces with preserved statistics of day-0 and day-95 (0.1st percentile is shown). Within-day angles were found between subspaces relative to even-numbered and odd-numbered trials of day-0 neural recordings. Principal angle values were averaged across four different time intervals (relative to initial decoder training) indicated by the transparency of the line (lighter for days closer to day-0, darker for days further away from day-0).

## Comparison of GANs to other methods for iBCI stabilization

Other approaches to address iBCI decoder instability include supervised techniques that aim at stabilizing iBCI performance by recalibrating the decoder during ongoing iBCI control by relying on access to the task output variables (*Dangi et al., 2014*; *Jarosiewicz et al., 2015*; *Orsborn et al., 2012*), as well as unsupervised methods that do not require to re-estimate decoder parameters and only need

neural data, with no provided task output variables or task labels (*Degenhart et al., 2020*; *Farshchian et al., 2018*; *Gallego et al., 2020*; *Karpowicz et al., 2022*; *Willett et al., 2021*). We restricted our comparison to GAN-based aligners and PAF for several reasons. First, both GANs and PAF are unsupervised methods. We argue that unsupervised methods are ideal for iBCI stabilization: because they do not require data labels, they should be simpler to implement in eventual clinical applications. Second, neither GANs nor PAF require trial alignment of the data, which CCA does require. This flexibility allowed us to align the neural data for more complicated behaviors. For example, one task in this study was a random-target reaching task in which monkeys moved a cursor between targets as they appeared on screen; this task structure produces movements of random length and direction, with varied speed and duration. Despite this complexity, all three of the tested aligners could still achieve good performance. Importantly, though, we previously demonstrated that ADAN still achieves higher performance than both CCA (*Gallego et al., 2020*) and KLDM (*Farshchian et al., 2018*) for the stereotyped isometric wrist task (*Farshchian et al., 2018*).

Although earlier attempts to achieve alignment via KLDM achieved only moderate success, a recent approach using KLD to align neural latent dynamics identified using Latent Factor Analysis through Dynamical Systems (LFADS) (*Pandarinath et al., 2018*; *Sussillo et al., 2016a*) was more successful (*Karpowicz et al., 2022*). Comparing this approach (called Nonlinear Manifold Alignment with Dynamics, or NoMAD) with Cycle-GAN turns out to be problematic because they are solving overlapping but different problems. A stable iBCI device has several interacting components: data preprocessing, an aligner that registers neural representations across days, and a decoder that translates neural activity to a predicted motor command. Higher iBCI performance could arise from an improvement to any of these processes. NoMAD includes the first two steps, performing both alignment of the neural representations via KLDM and data preprocessing via LFADS-based smoothing. Because Karpowicz et al., contrast NoMAD (alignment +powerful dynamics-based smoothing) to two methods that perform alignment with only very simple linear smoothing (ADAN and PAF), it is not possible to tell from their manuscript the extent to which NoMAD's higher performance arises from better alignment vs their use of LFADS for data smoothing. Nevertheless, the effects of the preprocessing can be inferred from their results: because of its more powerful dynamics preprocessing, NoMAD outperforms ADAN (and PAF) not only at day-k, but also on day-0 where neural alignment is not involved. The day-0 performance makes it clear that a substantial portion of NoMAD's higher performance comes not from its KLD-based alignment but from how the neural recordings are preprocessed with LFADS.

We can also draw conclusions purely from the method NoMAD uses for alignment, namely by minimizing the KLD between the distributions of day-0 and day-k states that come out of a day-0 LFADS Generator network. This alignment strategy is very similar to the KLDM method tested in *Farshchian et al., 2018*, where KLDM between neural states (obtained via an autoencoder) had inferior performance compared even to ADAN. This suggests that the apparent performance improvement of NoMAD over ADAN is a consequence of its embedded LFADS model rather than an indicator of KLD being a better alignment strategy. Theoretically, one could therefore replace the KLD-based alignment in NoMAD with a Cycle-GAN-based aligner and achieve even better performance. Going forward, it will be important for the field to establish consensus benchmark datasets and evaluation methods to disentangle the contributions of new methods in data preprocessing, neural alignment, and decoding, within each of these three areas.

A very different approach to iBCI stabilization was proposed by Sussillo et al., who trained a decoder with a large dataset spanning many months, under the hypothesis that neural turnover allows neurons not only to disappear, but potentially also to reappear later (*Sussillo et al., 2016b*). Although making the decoder robust to changes in the recorded neural populations, this approach has the inherent disadvantage of requiring the accumulation of a long stretch of historical data, which might be impractical for clinical use. In contrast to this approach, neither Cycle-GAN nor ADAN has a special requirement for the robustness of the day-0 decoder, and effective performance can be achieved with remarkably little data (*Figure 5*).

## iBCI stabilization without manifolds

CCA, KLDM, PAF, and ADAN all rely on dimensionality reduction of the recorded neural population prior to alignment. As a result, a portion of the variance of recorded neural activity is always

lost in the alignment process. In contrast, Cycle-GAN allows alignment to be performed on the full-dimensional neural recording, and achieves a superior performance compared to ADAN and PAF (*Figure 3*). This also means that Cycle-GAN can be used directly with any previously trained spike-rate based decoder. This is in contrast to ADAN and PAF, which only align the neural latent space and therefore require either a new, latent space decoder to be trained, or an additional post-alignment, backwards-projection step to convert the latent representation into a predicted set of spikes. The backwards-projection step leads to lower decoding performance for ADAN, and complete failure for PAF, as shown in *Figure 3—figure supplement 1*.

Because Cycle-GAN operates in the higher-dimensional space of the recorded neurons, it also recovers the response properties of individual neurons following alignment, providing the means to infer their response properties across many days of recording, even when those neurons are not actually observed. While single-neuron signals can in principle be generated by manifold-based alignment methods, we show here that these more indirectly reconstructed firing rates are less accurate (*Figure 6*). The potential applications of this ability to synthesize neural data from population recordings are yet undeveloped but intriguing. One possibility is that this strategy could be used to synthesize a "null distribution" of neural responses, to better detect effects of learning or behavioral changes that alter the response distribution of cells.

## Sources of decoding error following cross-day alignment

In this study, we relied on offline estimates of decoder accuracy, as they allowed us to examine large amounts of previously collected data across many monkeys and tasks. Also, by literally taking the monkey out of the loop, we were able to examine the accuracy of the alignment and decoding processes without the added complication of the monkeys' unknown and variable adaptation to the decoder. Although alignment by either ADAN or Cycle-GAN significantly improved the performance of a day-0 decoder on a given day-k, in most cases it did not attain the performance of a re-calibrated decoder, especially at long time offsets between day-0 and day-k (*Figure 3B*). One interesting potential cause of aligner performance drop is a change in the animal's behavioral strategy across days. Because the limb is kinematically redundant, the same hand position can be achieved with different limb postures (e.g. wrist angle) and muscle activation patterns. Similarly, differing strategies might be adopted to grasp the power or pinch force transducers. Even within a single experimental session, an M1 decoder trained on one behavior often fails to perform well when tested on a different behavior. Similarly, unsupervised M1 alignment will not be able to compensate for changes in strategy if they shift EMG (or kinematic) signals outside the space of values observed during training of the original decoder. We find some evidence for such drift in some tasks (predominantly the key grasp, *Figure 2—figure supplement 3C*), as indicated by differences between within- and across-day MMD of the motor outputs. Such differences were small, but could not be neglected (*Figure 2—figure supplements 2C and 3*).

## Network training challenges

Training GANs is a challenging task, in part because the learning rates of generator and discriminator networks must be carefully balanced to allow the networks to be trained in tandem (*Farnia and Ozdaglar, 2020*; *Salimans et al., 2016*). Many strategies have been proposed to improve the stability of learning and facilitate the convergence of GANs (*Arjovsky and Bottou, 2017*; *Brock et al., 2019*; *Farnia and Ozdaglar, 2020*; *Nagarajan and Kolter, 2017*; *Pan et al., 2019*; *Salimans et al., 2016*). ADAN and Cycle-GAN incorporate several of those strategies. First, both networks include an L1 loss term in their objective function, a modification that has been found in practice to improve the stabilization of model training by encouraging sparseness of model weights (*Arjovsky and Bottou, 2017*). The networks also use a two-timescale update rule for generator and discriminator learning rates, which facilitates convergence of generator and discriminator to a balanced solution (*Heusel et al., 2017*).

Correct optimization of GANs is also directly linked to proper tuning of the dynamics of learning during training (*Kurach et al., 2018*; *Saxena and Cao, 2021*), which we investigated here in depth. Given the many GAN variants, there are still no comprehensive guidelines for a particular architecture (*Ghosh et al., 2020*). Consistent with this, we found that ADAN and Cycle-GAN differ substantially in their sensitivity to learning rate and batch size hyperparameters. Notably, ADAN exhibited poor

generalization with larger batch sizes (like *Keskar et al., 2016*), while Cycle-GAN worked well across all tested values (*Figure 4A*). The ability to work with larger batch sizes gave Cycle-GAN several advantages over ADAN: its training was faster than ADAN (*Figure 4B*) and it also enabled Cycle-GAN to maintain stable performance with higher learning rates (*Figure 4C and D*, similar to the observations of *Goyal et al., 2017*).

## Conclusions

In summary, we demonstrated the successful use of GANs for the stabilization of an iBCI, thereby overcoming the need for daily supervised re-calibration. Both approaches we tested (ADAN and Cycle-GAN) require remarkably little training data, making them practical for long-term iBCI clinical applications. Between the two approaches, Cycle-GAN achieved better performance which was less affected by inaccurate hyperparameter tuning; it is therefore our recommended method for future use. Notably, Cycle-GAN works directly with the unstable full-dimensional neural recordings, which further increases its performance and simplifies its implementation.

# Materials and methods

## Subjects and behavior tasks

Six 9–10 kg adult male rhesus monkeys (Macaca mulatta) were used in this study. They were trained to sit in a primate chair and control a cursor on a screen in front of them using different behavioral apparatuses (*Figure 2—figure supplement 1*).

Monkeys J and S were trained to perform an isometric wrist task, which required them to control the cursor on the screen by exerting forces on a small box placed around one of the hands. The box was padded to comfortably constrain the monkey's hand and minimize its movement within the box, and the forces were measured by a 6 DOF load cell (JR3 Inc, CA) aligned to the wrist joint. During the task, flexion/extension force moved the cursor right and left respectively, while force along the radial/ulnar deviation axis moved the cursor up and down. Each trial started with the appearance of a center target requiring the monkeys to hold for a random time (0.2–1.0 s), after which one of eight possible outer targets selected in a block-randomized fashion appeared, accompanied with an auditory go cue. The monkey was allowed to move the cursor to the target within 2.0 s and hold for 0.8 s to receive a liquid reward. For both decoding and alignment analyses, we only used the data within each single trial (from 'trial start' to 'trial end', *Figure 2—figure supplement 1A*). We did not do any temporal alignment with the trials, so the lengths of the trials were different from each other.

Monkeys P and G were trained to perform a grasping task, which required them to reach and grasp a gadget placed under the screen with one hand. The gadget was a cylinder for monkey P facilitating a power grasp with the palm and the fingers, while a small rectangular cuboid for monkey G facilitating a key grasp with the thumb and the index finger. A pair of force sensitive resistors (FSRs) were attached on the sides of the gadgets to measure the grasping forces the monkeys applied. The sum and the difference of the FSR outputs were used to determine the position of the cursor on the vertical axis and the horizontal axis respectively. At the beginning of each trial the monkey was required to keep the hand resting on a touch pad for a random time (0.5–1.0 s). A successful holding triggered the onset of one of three possible rectangular targets on the screen and an auditory go cue. The monkey was required to place the cursor into the target and hold for 0.6 s by increasing and maintaining the grasping force applied on the gadget (*Figure 2—figure supplement 1B*). For this task we extracted trials from 'go cue time' to 'trial end', as the monkeys' movements were quite random before the go cue.

Monkeys C and M were trained to perform a center-out (CO) reaching task while grasping the upright handle of a planar manipulandum, operated with the upper arm in a parasagittal plane. Monkey C performed the task with the right hand, monkey M with the left. At the beginning of each trial the monkey needed to move the hand to the center of the workspace. One of eight possible outer targets equally spaced in a circle was presented to the monkey after a random waiting period. The monkey needed to keep holding for a variable delay period until receiving an auditory go cue. To receive a liquid reward, the monkey was required to reach the outer target within 1.0 s and hold within the target for 0.5 s (*Figure 2—figure supplement 1C*). For this task we extracted trials from 'go cue time' to 'trial end', since the monkeys kept static before the go cue.

Monkey M was trained to perform a random-target (RT) task, reaching a sequence of three targets presented in random locations on the screen to complete a single trial. The RT task used the same apparatus as the CO reach task. At the beginning of each trial the monkey also needed to move the hand to the center of the workspace. Three targets were then presented to the monkey sequentially, and the monkey was required to move the cursor into each of them within 2.0 s after viewing each target. The positions of these targets were randomly selected, thus the cursor trajectory for each trial presented a 'random-target' manner (*Figure 2—figure supplement 1D*). For this task we extracted trials from 'trial start' to 'trial end'.

All surgical and experimental procedures were approved by the Institutional Animal Care and Use Committee (IACUC) of Northwestern University under protocol #IS00000367, and are consistent with the Guide for the Care and Use of Laboratory Animals.

## Implants and data recordings

Depending on the task, we implanted a 96-channel Utah electrode array (Blackrock Neurotech, Inc) in either the hand or arm representation area of the primary motor cortex (M1), contralateral to the arm being used for the task (see *Figure 2—source data 1*). The implant site was pre-planned and finally determined during the surgery with reference to the sulcal patterns and the muscle contractions evoked by intraoperative surface cortical stimulation. For each of monkeys J, S, G, and P, we also implanted intramuscular leads in forearm and hand muscles of the arm used for the task in a separate procedure (see *Figure 2—source data 1*). Electrode locations were verified during surgery by stimulating each lead.

M1 activity was recorded during task performance using a Cerebus system (Blackrock Neurotech, Inc). The signals on each channel were digitalized, bandpass filtered (250~5000 Hz) and converted to spike times based on threshold crossings. The threshold was set with respect to the root-mean square (RMS) activity on each channel and kept consistent across different recording sessions (monkeys J, C and M: –5.5 x RMS; monkey S: –6.25 x RMS; monkey P: –4.75 x RMS; monkey G: –5.25 x RMS). The time stamp and a 1.6 ms snippet of each spike surrounding the time of threshold crossing were recorded. For all analyses in this study, we used multiunit threshold crossings on each channel instead of discriminating well isolated single units. We applied a Gaussian kernel (S.D.=100 ms) to the spike counts in 50 ms, non-overlapping bins to obtain a smoothed estimate of firing rate as function of time for each channel.

The EMG signals were differentially amplified, band-pass filtered (4-pole, 50~500 Hz) and sampled at 2000 Hz. The EMGs were subsequently digitally rectified and low-pass filtered (4-pole, 10 Hz, Butterworth) and subsampled to 20 Hz. EMG channels with substantial noise were not included in the analyses, and data points of each channel were clipped to be no larger than the mean plus 6 times the S.D. of that channel. Within each recording session, we removed the baseline of each EMG channel by subtracting the 2nd percentile of the amplitudes and normalized each channel to the 90th percentile. For monkeys C and M, we recorded the positions of the endpoint of the reach manipulandum at a sampling frequency of 1000 Hz using encoders in the two joints of the manipulandum.

## iBCI day-0 decoder

The day-0 decoder was a Wiener filter of the type that we have used in several previous studies (*Cherian et al., 2011*; *Naufel et al., 2019*). The filter was fit using linear regression to predict the motor outputs (either EMG or hand velocity) at time $t$ given neural responses from time $t$ to time $t$ - T, where we set T=4 (200 ms) for all decoders used in this study. As the aligners being tested worked with either low-dimensional manifolds or the full neural population, and required the associated day-0 decoders to be compatible, we implemented different day-0 decoders to match the outputs of the aligners. For Cycle-GAN, we trained a Wiener filter using the full-dimensional neural firing rates recorded on day-0. For ADAN and PAF, we performed dimensionality reduction (ADAN: autoencoder, PAF: Factor Analysis; dimensionality = 10 for both) to find a low-dimensional latent space, and trained the decoder using the projections of the neural signals into this latent space. The Wiener filters were trained using the day-0 data with four-fold cross validation, and the filter corresponding to the fold with the best $R^2$ was selected as the fixed day-0 decoder. The parameters for the dimensionality reduction procedures and the Wiener filter from the day-0 data were kept fixed for decoding on subsequent days.

## iBCI aligners

### Adversarial domain adaptation network (ADAN)

We adhered to the main architecture and the training procedures of the ADAN as described in *Farshchian et al., 2018*. Briefly, we first find a nonlinear latent space by jointly training an autoencoder and a long short-term memory (LSTM) neural network-based iBCI decoder using day-0 data. (Note that this LSTM based decoder is only used for latent space discovery, not the later decoding stage that is used for performance evaluation (see 'ADAN day-0 training' in **Appendix** for full details)). We then construct an adversarial aligner comprised of a distribution alignment module (generator network G) and a discriminator network D (*Appendix 1—figure 1*), where G is a shallow feedforward neural network, and D is an autoencoder with the same architecture as that used for the day-0 latent space discovery. During training of the aligner, G is fed with day-k neural firing rates and applies a nonlinear transform over these data to match them to the day-0 neuron response distributions. The output of G, and the true day-0 neural firing rates are then passed to D, which passes both inputs through the autoencoder: namely, it projects each signal into the latent space and then reconstructs it. The distributions of the residuals between the autoencoder inputs and the reconstructions are computed for both the generator output and the true day-0 data, and a lower bound to the Wasserstein distance is used to measure the dissimilarity between the two distributions. The goal of adversarial learning is to find a discriminator D that maximizes the dissimilarity between responses of D to true day-0 firing rates and to outputs of G, while also finding a generator G that minimizes the dissimilarity between true day-0 firing rates and the outputs of G; this objective is called the adversarial loss. When the training is completed, G will have been trained to 'align' the neural firing rates on day-k with those on day-0. For a full description of the ADAN architecture and its training strategy, please refer to **Appendix** and (*Farshchian et al., 2018*).

### Cycle-GAN

The Cycle-GAN aligner is based on the structure proposed in *Zhu et al., 2017*. Like ADAN, Cycle-GAN does not consider any dynamic information, aligning only the point clouds representing the instantaneous firing rate of M1 neurons. Unlike ADAN, it converts the full-dimensional neural firing rates collected on day-k into a form resembling those collected on day-0, with no dimensionality reduction. Cycle-GAN consists of two feedforward generator neural networks ($G_1$ and $G_2$) and two discriminator networks ($D_1$ and $D_2$, see *Appendix—figure 1B*). These form two pairs of adversarial networks: $G_1$ maps data from the day-k domain to the day-0 domain, while $D_1$ aims to distinguish between the day-0 samples and the output of $G_1$. And in parallel, $G_2$ maps data in the day-0 domain to the day-k domain, while $D_2$ distinguishes day-k data from output of $G_2$. In contrast to ADAN, the cycle-GAN discriminator networks operate directly on neural responses, rather than the residuals between low-dimensional and full-dimensional responses.

The objective function for network training has two major terms. The first is an adversarial loss, defined for both generator-discriminator pairs ($G_1 + D_1$ and $G_2 + D_2$) as in ADAN. The second term is known as the cycle-consistency loss, which pushes the mappings $G_1$ and $G_2$ to become inverses of each other: that is, a sample from one specific domain should be recovered to its original form after going through the cycle composed of the two mappings. As argued by Zhu et al, the introduction of the cycle-consistency loss regularizes the learning of the mapping functions, thereby reducing the search space. In (*Appendix—figure 1B*) the purple arrows through $G_1$ and $G_2$ reflect the transformation of each sample from the day-k domain into the day-0 domain by $G_1$, followed by the recovery from the day-0 domain into the day-k domain by $G_2$. Likewise, the orange arrows through $G_2$ and $G_1$ reflect a transformation from the day-0 domain to the day-k domain and back to the day-0 domain. Further details about the Cycle-GAN based aligner are provided in **Appendix**.

### GAN training and architecture

Both ADAN and Cycle-GAN were trained using the ADAM optimizer (*Kingma and Ba, 2015*) with a four-fold cross validation. We used 400 training epochs and reported the alignment result that produced the best decoder performance on a held-out validation set of trials. In addition to the learning hyperparameters explored in the Results section, we examined several different architectures for the aligner neural network of both ADAN and Cycle-GAN (varying the number of layers and neurons per layer), and replaced the least absolute deviations (L1) for both the adversarial and

cycle-consistency loss with the least square error (L2) (*Mao et al., 2016*). None of the manipulations substantially improved performance.

## Procrustes alignment of factors (PAF)

We compared ADAN and Cycle-GAN aligners with a manifold-based stabilization method proposed by *Degenhart et al., 2020*, the Procrustes Alignment of Factors (PAF, our term). PAF finds a low-dimensional manifold using Factor Analysis, then applies a Procrustes transformation to the neural manifold of day-0 to align it to that of day-k. The original application of PAF additionally removes electrodes identified as "unstable" and unlikely to contribute to alignment; these are defined as electrodes on day-k that have changed the most with respect to the day-0 manifold, and are removed iteratively until a criterion is met. However, we found that alignment performance did not degrade with the number of included electrodes, so we decided to omit this stability criterion and use all recorded electrodes for all the datasets. As for the GAN aligners, we trained and tested PAF using a Wiener filter and four-fold cross validation.

## Performance measures

### Decoder accuracy

To evaluate the performance of decoders mapping M1 neural recordings to motor outputs (either EMG or hand velocity), we used the coefficient of determination ($R^2$). The $R^2$ indicates the proportion of variation of the actual motor output that was predicted by the iBCI decoder; this approach is common in evaluation of iBCI systems (*Morrow and Miller, 2003*). As the motor outputs being decoded are multi-dimensional (7 dimensions for EMG, 2 dimensions for hand velocity), we computed a multivariate $R^2$ in which, after computing the $R^2$ for all the single dimensions, we take a weighted average across dimensions, with weights determined by the variance of each dimension. This was implemented using the 'r2_score' function of the scikit-learn python package with 'variance weighted' for the 'multioutput' parameter (*Pedregosa et al., 2011*).

### Maximum mean discrepancy (MMD)

We used maximum mean discrepancy (MMD) in two contexts. First, we used MMD to evaluate the similarity between the distribution of the aligned day-k neural activity and the day-0 neural activity, as a way to examine the alignment performance (*Figure 6*). MMD provides a measure of distance between two multivariate distributions, based on the distances between the mean embeddings of samples drawn from each distribution in a reproducing kernel Hilbert space (*Gretton et al., 2012a*). MMD is symmetric in the two distributions and equals zero if and only if the two distributions are the same. To select our kernel, we followed a technique that has been proved feasible for optimizing kernel choice (*Gretton et al., 2012b*): specifically we employed a family of four Gaussian kernels with width between 5 Hz and 50 Hz. To define a 'smallest possible' MMD between aligned day-k and day-0 distributions, we divided neural signals recorded on the same day into non-overlapping folds, and computed MMD between them; we call this the 'within-session MMD' in *Figure 6*.

We also use the MMD to quantify the similarity of the distributions of neural activity or motor outputs between pairs of separate recording sessions for each dataset, as a way to quantify the recordings instabilities (*Figure 2—figure supplements 2C and 3*). For a pair of sessions, we divided each of them into four non-overlapping folds, and computed the MMD between each fold and its counterpart in the other session, then reported the mean value across folds. We also computed the 'within-session MMD' for neural activity/motor outputs for each session, using the same way described above.

### Principal angles

To evaluate the similarity between neural manifolds of day-0 and day-k before and after alignment, we used principal angles (*Knyazev and Argentati, 2002*). Principal angles provide a metric to quantify the alignment of two subspaces embedded in a higher-dimensional space. For any pair of *C*-dimensional hyperplanes, there are *C* principal angles that exist between them. Following the approach outlined in *Knyazev and Argentati, 2002* and *Elsayed et al., 2016*, these angles are computed as follows: first, we reduce each signal (here the day-0 and day-k neuron firing rates) to 10 dimensions using PCA. Next, recursively for each C=1...10, we identify the pair of principal vectors that are

separated by the smallest angle and that are also perpendicular to the prior selected pairs, and report that angle. When two hyperplanes are well-aligned, the leading principal angles between them can be very small, but often the last few angles are quite large. We computed the principal angles using the 'subspace_angles' function of the SciPy python package (*Virtanen et al., 2020*).

To assess whether the angles after neural alignment were significantly small, we compared them to an upper bound provided by the angle between two surrogate subspaces, using the strategy described in *Elsayed et al., 2016*. Briefly, we generated 10,000 random pairs of day-0-like and day-95-like subspaces in which we shuffled the timing of spikes within each neuron, destroying correlation structure while preserving the statistics of neural firing rates within each day. We then computed the principal angles between each pair, and used the 0.1th percentile of the principal angle distribution as the threshold below which angles could be considered smaller than expected by chance given firing rate statistics alone. We also defined a 'within-day' bound by computing the principal angles between the day-0 neural recordings of even-numbered and odd-numbered trials, to reduce to a minimum the effect of any within-day drift. If the alignment process is successful, we expect the neural manifolds of day-0 and day-k to have principal angles similar to those of the within-day bound.

## Statistics

We applied statistical tests to compare the decoding accuracy over time after neural alignment with Cycle-GAN, ADAN, and PAF. For these comparisons, we ran a linear mixed-effect model with the type of aligner and the number of days elapsed from decoder training as fixed factors and the type of task as a random factor. In addition, we compared the performance of Cycle-GAN and ADAN with different hyperparameter settings, including generator and discriminator learning rates, as well as batch size. For all these comparisons, we used a two-sided Wilcoxon's signed rank test. We also used a two-sided Wilcoxon's signed rank to test whether there was a significant difference between any two methods when limited amount of training data was used for alignment. Finally, we compared the MMD of neural distributions between all pairs of day-0/day-k sessions before and after alignment with Cycle-GAN and ADAN. Since the distributions pre and after alignment are independent, we used a two-sided Wilcoxon's rank sum test. For all the statistical models, we used a significance threshold of $\alpha$=0.05. When making pairwise comparisons between the three aligners, we used a Bonferroni correction of 3. Sample sizes are reported in the corresponding results section.

## Acknowledgements

We thank Ali Farshchian, Sara Solla and Ege Altan for valuable discussions. We thank current and former members of the Miller Limb Lab, including Stephanie Naufel, Matthew Perich, and Christian Ethier, for their contributions to data collection. The work was supported in part by grants to LEM (R01 NS053603, R01 NS074044).

## Additional information

### Funding

| Funder | Grant reference number | Author |
| --- | --- | --- |
| National Institute of Neurological Disorders and Stroke | R01 NS053603 | Lee E Miller |
| National Institute of Neurological Disorders and Stroke | R01 NS074044 | Lee E Miller |

The funders had no role in study design, data collection and interpretation, or the decision to submit the work for publication.

### Author contributions

Xuan Ma, Fabio Rizzoglio, Conceptualization, Data curation, Investigation, Visualization, Methodology, Writing – original draft, Writing – review and editing; Kevin L Bodkin, Data curation, Investigation; Eric

Perreault, Conceptualization, Supervision, Writing – review and editing; Lee E Miller, Conceptualization, Supervision, Funding acquisition, Writing – original draft, Writing – review and editing; Ann Kennedy, Conceptualization, Supervision, Writing – original draft, Writing – review and editing

### Author ORCIDs
Xuan Ma (ID) https://orcid.org/0000-0003-3352-1905
Fabio Rizzoglio (ID) http://orcid.org/0000-0002-6744-4605
Lee E Miller (ID) http://orcid.org/0000-0001-8675-7140
Ann Kennedy (ID) http://orcid.org/0000-0002-3782-0518

### Ethics
All surgical and experimental procedures were approved by the Institutional Animal Care and Use Committee (IACUC) of Northwestern University under protocol #IS00000367, and are consistent with the Guide for the Care and Use of Laboratory Animals.

### Decision letter and Author response
Decision letter https://doi.org/10.7554/eLife.84296.sa1
Author response https://doi.org/10.7554/eLife.84296.sa2

## Additional files

### Supplementary files
• MDAR checklist

### Data availability
Data from all animals and tasks is available via Dryad at: https://doi.org/10.5061/dryad.cvdncjt7n.

The following dataset was generated:

| Author(s) | Year | Dataset title | Dataset URL | Database and Identifier |
|---|---|---|---|---|
| Ma X, Rizzoglio F, Thacker S, Miller L | 2023 | Using adversarial networks to extend brain computer interface decoding accuracy over time | https://doi.org/10.5061/dryad.cvdncjt7n | Dryad Digital Repository, 10.5061/dryad.cvdncjt7n |

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

## Appendix 1

## Detailed methods for iBCI decoders and aligners

Testing neural alignment on your data

We provide a step-by-step tutorial on the use of CycleGAN and ADAN for neural alignment on GitHub in our adversarial_BCI repository: https://github.com/limblab/adversarial_BCI, (copy archived at swh:1:rev:187857d4963dcffbdbf633502b1e41dafa4cd09a; *Ma, 2023a*) in the Jupyter notebooks ADAN_aligner.ipynb and Cycle_GAN_aligner.ipynb. Briefly, the steps covered by these notebooks are as follows:

1. Set up requirements. In addition to common Python data science libraries, our alignment code makes use of the following more specialized packages:
   a. XDS cross-platform data structure, documentation for which can be found at https://github.com/limblab/XDS, (copy archived at swh:1:rev:104719352b92cfa9200f2d-d91902151295aceea9; *Ma, 2023b*). Datasets should be packaged into the XDS format for analysis using the provided notebooks, or else datasets should be formatted into lists of numpy arrays as described in the notebook (see documentation on variables day0_spike and day0_EMG in the notebook).
   b. A simple Wiener filter decoder module, found at https://github.com/xuanma/decoder_standard, (copy archived at swh:1:rev:032a8491381a9ac9267b0bd8003d84c10743aa35; *Ma, 2023c*).
   c. Pytorch, a Python library for working with deep neural network models, is required for Cycle-GAN. Tensorflow 1.* is required for ADAN. Note that because our alignment models are quick to train, they do not require a computer with a GPU.
2. Data preprocessing. Given extracellular spike trains from an implanted recording device (here a 96-channel Utah array) we compute spike counts per channel using 50 ms time binning, then smooth these spike counts using a Gaussian kernel with a standard deviation of 100 ms. Our provided notebook loads and pre-processes neural recording data from two days of experiments, namely the source and target days. Similarly, raw EMG recordings are pre-processed via rectification and filtering, as described in the Methods section of this manuscript.
3. Trial selection. In our demo notebook, we use only the first 160 trials on a given recording day. Selection of trials is achieved by indexing into our day0_spike, day0_EMG, dayk_spike, and dayk_EMG lists, which contain the now-preprocessed data following spike count smoothing and EMG envelope extraction, segmented into trials.
4. Train the day-0 decoders. The goal of alignment is to be able to use a previously trained neural decoder to predict EMG activity on neural recording data from a new experimental session. In our work, we use a simple Wiener filter decoder as our "previously trained decoder". In the provided notebook, we explain the design of the Wiener filter decoder, and provide a function train_wiener_filter to train a Wiener filter decoder on the day-0 data as well as wrapper code implementing four-fold cross-validation. The training code applies data splits, formats data for the decoder, and trains and tests the Wiener filter decoder for each split, reporting back multivariate $R^2$ values and saving the best-performing decoder to a .npy file for further use. The notebook also uses a function plot_actual_and_pred_EMG to plot the predicted EMG signals using the decoder alongside the corresponding ground-truth EMG signals.
5. Define the blocks for the Cycle-GAN (or ADAN) aligner. The next step is to define the architecture of the alignment model. Model definition code is provided in both notebooks; for example, they Cycle-GAN notebook defines Generator and Discriminator classes, each of which has an __init__ function to initialize the network architecture and a function forward which takes an input firing rate signal and returns a transformed version of that signal.
6. Train the Cycle-GAN (or ADAN) aligner. Having designed the architecture of our model, we next provide a function train_cycle_gan_aligner to carry out training. This function first carries out several setup steps:
   a. Specifying the value of model hyperparameters (which can be set by the user).
   b. Initializing two Generator and two Discriminator networks.
   c. Defining the type of loss function (MSE or L1-penalized) and optimizer to be used by the model; these are standard terms provided by torch.

d. Initializing DataLoader objects to feed the training or test dataset into the torch model. It then carries out the GAN training loop. Briefly, each iteration does the following, where "loss" is by default the mean-squared error between two signals:

e. Sample a pair of neural recording trials, one from day-0 and one from day-k.

f. Compute the identity loss, which takes the error between day-k data and its transformation by the day-0 Generator (and similarly for day-0 and the day-k Generator). This loss regularizes the Generator to be close to an identity mapping when provided with samples from its target domain, an approach used in the original Cycle-GAN manuscript and adopted from Taigman, Polyak, & Wolf 2017.

g. Compute the GAN loss for the day-0 data using the day-0 Generator + Discriminator (and similarly for day-k). For the day-0 Generator, this loss is the accuracy of the Discriminator in distinguishing true day-k data from synthetic day-k data; for the day-0 Discriminator, the loss is the error rather than the accuracy.

h. Compute the cycle-consistency loss for the day-0 data by feeding the synthetic day-k data through the day-k Generator and computing the error between this output and the original day-0 data (and similarly for day-k).

i. Sum applicable losses for each Generator (identity, GAN-Generator, and cycle-consistency) and each Discriminator (GAN-Discriminator only).

j. Compute the gradient with respect to each loss, and pass this information to the optimizer, which will update the model parameters at the end of each epoch.

k. To monitor training progress, the aligner is evaluated on the validation set every 10 epochs, and performance is logged.

7. Test the trained aligner. The provided function test_cycle_gan_aligner takes as input a trained aligner and a neural dataset, and returns as output the aligned version of that dataset. It does this by passing the data through the trained model network.

8. Plot performance. The notebook next shows how to evaluate the quality of the previously trained day-0 decoder when fed aligned neural signals. As in step 3 above, the provided function plot_actual_and_pred_EMG is used, but now we are feeding aligned day-k neural activity into the decoder and comparing the decoder's prediction to the day-0 EMG.

We advise the reader to consult the complete Jupyter notebook for additional commentary and documentation of these steps. In addition to this practical guide to use of Cycle-GAN for alignment, we have included additional technical documentation of the alignment process in the following sections.

## iBCI day-0 decoders

We used a Wiener filter (*Cherian et al., 2011*) as the day-0 iBCI decoder:

$$y\left(t\right) = \sum_{\tau=0}^{T-1} \beta\left(\tau\right) x\left(t - \tau\right) \tag{1}$$

where $y\left(t\right)$ is a q-dimensional vector (q is 2 for hand velocity prediction and varied with the number of recorded EMGs for EMG prediction, see *Appendix 1—table 1*) representing the motor outputs to be predicted at time $t$, while $x\left(t\right)$ is a p-dimensional vector for the inputs to the Wiener filter at time $t$, and $\beta\left(\tau\right)$ is a q × p matrix corresponding to the filter parameters for time step $\tau$. For Cycle-GAN, $x\left(t\right)$ is the full-dimensional neural firing rates, thus p equals to the number of the electrodes in the cortical array (denoted as C). For ADAN, $x\left(t\right)$ is the projection of the neural firing rates in a nonlinear latent space found by an autoencoder (see next section for details). For PAF, $x\left(t\right)$ is the projection of the neural firing rates in a linear latent space found by factor analysis. We set p = 10 for both ADAN and PAF. We can also write *Equation 1* in matrix form:

$$Y = XB \tag{2}$$

where $Y$ is a M × q matrix for the motor outputs to be predicted with M being the number of samples, $X$ is a M × (T × p) matrix, and $B$ is a (T × p) × q matrix for the regression coefficients to be estimated. We also added an additional bias term for both $X$ and $B$. $B$ was determined by a ridge regression estimator:

$$\hat{B} = \left(X^{\mathrm{T}} X + \lambda I\right)^{-1} X^{\mathrm{T}} Y \tag{3}$$

We chose a ridge regression to limit the risk of decoder overfitting by penalizing solutions with large regression coefficients with the regularization term $\lambda$. The value of $\lambda$ was chosen by sweeping a range of 20 values between 10 and $10^5$ on a logarithmic scale. We used a 4-fold cross validation to train the decoder for each aligner type and ultimately selected the model with the highest $R^2$ on the test set as the fixed day-0 decoder.

## ADAN day-0 training

The day-0 wiener filter for ADAN was built from a nonlinear latent space estimated from day-0 neural firing rates using an autoencoder (AE) originally described in *Farshchian et al., 2018*. The AE consists of an input layer, five hidden layers and an output layer. The input and the output layers have C units, while the hidden layers (from input to output) have 64, 32, 10, 32 and 64 units, respectively. Hence, the AE compresses the C-dimensional neural firing rates into a 10-dimensional latent representation. The units in the layer and the output layers as well as those in the latent layer have linear activation functions, while units in the remaining hidden layers have a nonlinear one (exponential linear unit, ELU). The AE is trained to minimize the reconstruction error defined as the mean square error (MSE) between the input and the output data. When day-0 neural firing rates $\{x\}$ are fed through the AE, the latent layer activity $\{l\}$ and the corresponding reconstructions $\{\hat{x}\}$ are obtained. The 10-dimensional latent activity $\{l\}$ is then mapped onto the q-dimensional motor output vector through a long-short-term memory (LSTM, *Hochreiter and Schmidhuber, 1997*):

$$\hat{y} = \mathrm{LSTM}\left(l\right) \tag{4}$$

where $y$ is the actual motor output (either EMG or hand velocity) recorded at day-0 and $\hat{y}$ is its prediction with the LSTM. The LSTM is designed with one layer and a number of units that equals the number of recorded EMGs (if the motor output is EMG) or two (if the motor output is hand velocity). The AE and the LSTM are simultaneously trained by minimizing a loss function that accounts for both the MSE of the reconstruction of the firing rates ($\mathcal{L}(\mathrm{AE})$) and the MSE of the motor output predictions ($\mathcal{L}(\mathrm{LSTM})$):

$$\mathcal{L} = \lambda \mathcal{L}(\mathrm{AE}) + \mathcal{L}(\mathrm{LSTM}) = \frac{1}{\mathrm{M}} \sum_{i=1}^{\mathrm{M}} \left(\lambda \|\hat{x} - x\|^2 + \|\hat{y} - y\|^2\right) \tag{5}$$

where M is the total number of training samples. The weighting factor $\lambda$ equalizes the contribution of the two terms so that the learning algorithm does not prioritize one over the other. For each training epoch, $\lambda$ is updated as the ratio between the values of $\mathcal{L}(\mathrm{AE})$ and $\mathcal{L}(\mathrm{LSTM})$ at the end of the preceding epoch.

The simultaneous training of the AE and the LSTM allows extracting a low-dimensional space of neural activity constrained to include features related to movement intent. Such neural manifold is then used to train the Wiener filter used as the fixed day-0 decoder for this study. At each epoch of training, the current latent signal $\{l\}$ was used as input for *Equation 3* to obtain a linear prediction of the actual motor output. We used 400 epochs of training and ultimately selected the parameters of the wiener filter at the epoch that had the best performance (in the $R^2$ sense) on the held-out test set.

ADAN based aligner. The discriminator D of ADAN is an autoencoder (*Appendix 1—figure 1A*), and has the same architecture as that used to find the nonlinear latent space on day-0 (day-0 AE). The parameters of D ($\theta_{\mathrm{D}}$) are initialized with the parameters of the day-0 AE. The generator G is a feedforward neural network with one hidden layer with C neurons (i.e., the number of the electrodes in the cortical array). The parameters of G ($\theta_{\mathrm{G}}$) are initialized as identity matrices. We set a nonlinear activation function (ELU) for the hidden layer, and a linear one for the output layer.

Here we denote the day-0 neural firing rates as $\{x_i\}_{i=1}^{\mathrm{M}}$ and the day-k neural firing rates as $\{z_j\}_{j=1}^{\mathrm{N}}$, where both $x_i$ and $z_j$ are C-dimensional vectors representing the neural firing rates from C electrodes at a given time bin, and M and N are the total number of samples for day-0 and day-k data respectively. Since at one time we fed the networks with S training samples as a batch, we can write a training batch from $\{x\}$ or $\{z\}$ in matrix form as $X$ or $Z$. During training, we fed $Z$ to G and got G($Z$) as the aligned day-k neural firing rates. At the same time, we fed D with both G($Z$) and $X$. As D is an autoencoder, it would produce the reconstructions of them from the latent space, which

can be written as $\widehat{G(Z)}$ and $\hat{X}$. Hence, we could get the residuals between the true data and these reconstructions by computing:

$$
\begin{aligned}
\boldsymbol{R}_X &= X - \hat{X} \\
\boldsymbol{R}_{\text{G}(Z)} &= \text{G}\left(Z\right) - \widehat{\text{G}\left(Z\right)}
\end{aligned}
\tag{6}
$$

$\boldsymbol{R}_X$ and $\boldsymbol{R}_{\text{G}(Z)}$ are both $\text{S} \times \text{C}$ matrices. We then computed the scalar reconstruction losses as the $\text{L}_1$ norm of each column of $\boldsymbol{R}_X$ and $\boldsymbol{R}_{\text{G}(Z)}$. Let $\rho(\boldsymbol{R}_X)$ and $\rho(\boldsymbol{R}_{\text{G}(Z)})$ represent the distributions of these scalar losses, and let $\mu(\boldsymbol{R}_X)$ and $\mu(\boldsymbol{R}_{\text{G}(Z)})$ be the corresponding means of $\rho(\boldsymbol{R}_X)$ and $\rho(\boldsymbol{R}_{\text{G}(Z)})$. We measured the dissimilarity between $\rho(\boldsymbol{R}_X)$ and $\rho(\boldsymbol{R}_{\text{G}(Z)})$ by a lower bound to the Wasserstein distance (*Arjovsky et al., 2017*), which is given by the absolute value of the difference between $\mu(\boldsymbol{R}_X)$ and $\mu(\boldsymbol{R}_{\text{G}(Z)})$: $\text{W}(\rho(\boldsymbol{R}_X), \rho(\boldsymbol{R}_{\text{G}(Z)})) \geq |\mu(\boldsymbol{R}_X) - \mu(\boldsymbol{R}_{\text{G}(Z)})|$. The parameters of the generator ($\theta_\text{G}$) and discriminator ($\theta_\text{D}$) are updated via batch gradient descent by minimizing their corresponding cost functions:

$$
\begin{aligned}
\mathcal{L}\left(\text{D}\right) &= \mu\left(\boldsymbol{R}_X\right) - \mu\left(\boldsymbol{R}_{\text{G}(Z)}\right) \\
\mathcal{L}\left(\text{G}\right) &= \mu\left(\boldsymbol{R}_{\text{G}(Z)}\right)
\end{aligned}
\tag{7}
$$

For each epoch of training, $\mathcal{L}(\text{G})$ is first minimized and followed by $\mathcal{L}(\text{D})$. Minimizing $\mathcal{L}(\text{G})$ implies bringing the output of the generator (*i.e.*, the aligned day-k neural data, G($\boldsymbol{Z}$)) close to the day-0 data $\boldsymbol{X}$. When G($\boldsymbol{Z}$) is fed through D, residuals with mean $\mu_\boldsymbol{Z}$ are obtained. Since D is initialized with the day-0 AE weights, $\mu_\boldsymbol{Z}$ can be reduced if $\theta_\text{G}$ are updated to appropriately modify G($\boldsymbol{Z}$) and make it resemble $\boldsymbol{X}$. When $\mathcal{L}(\text{G})$ is minimized, the gradients flow through both D and G, but only the parameters $\theta_\text{G}$ are updated at this stage.

While G is trying to decrease $\mu_\boldsymbol{Z}$, D is working as an adversary. Minimizing $\mathcal{L}(\text{D})$ implies maximizing the difference between $\mu(\boldsymbol{R}_X)$ and $\mu(\boldsymbol{R}_{\text{G}(Z)})$ (*i.e.*, their Wasserstein distance W). Again, since D is initialized with the day-0 AE weights (and the generator is an identity matrix when training begins), the residuals of the day-k data will be greater than those of the day-0 data, hence $\left(\mu_\boldsymbol{Z} > \mu_\boldsymbol{X}\right)$. Thus, if $\theta_\text{D}$ are updated to maximize $\left(\mu_\boldsymbol{Z} - \mu_\boldsymbol{X}\right)$, or equivalently minimize $\left(\mu_\boldsymbol{X} - \mu_\boldsymbol{Z}\right)$, this relation is maintained during training. Since scalar residuals and their means are always nonnegative, maximization of W is achieved by decreasing $\mu_\boldsymbol{X}$ while increasing $\mu_\boldsymbol{Z}$. The adversarial mechanism between G and D ensures that the neural alignment is achieved in an unsupervised manner.

**Appendix 1—table 1.** ADAN hyperparameters.

| parameter | value |
| --- | --- |
| Total number of trainable parameters | 35,946 |
| Batch size | 8 |
| Discriminator (**D**) learning rate | 0.00005 |
| Generator (**G**) learning rate | 0.0001 |
| Number of training epochs | 200 |

## Cycle-GAN based aligner

The Cycle-GAN generators, $\text{G}_1$ and $\text{G}_2$ are both shallow feedforward neural networks with one hidden layer with C neurons. We set a nonlinear activation function (RELU) for the hidden layer, and a linear one for the output layer. The discriminators, $\text{D}_1$ and $\text{D}_2$ are also shallow feedforward neural networks with one hidden layer. The input layer and the hidden layer both have C neurons, while the output layer has 1 neuron, as the output is a class label indicating which distribution the input sample belongs to. Same as $\text{G}_1$ and $\text{G}_2$, the hidden layer of $\text{D}_1$ and $\text{D}_2$ uses a nonlinear activation function (RELU), and the output layer uses a linear one. The layer weights of each network were initialized through Xavier initialization.

As shown in (*Appendix 1—figure 1B*), we fed the day-k neural firing rates $\boldsymbol{Z}$ to $\text{G}_1$ to get the aligned day-k neural firing rates (G$_1$($\boldsymbol{Z}$)), and the day-0 neural firing rates $\boldsymbol{X}$ to $\text{G}_2$ to convert data in the day-0 domain back into the day-k domain (G$_2$($\boldsymbol{X}$)). Meanwhile, the discriminator $\text{D}_1$ was fed with $\boldsymbol{X}$ and (G$_1$($\boldsymbol{Z}$)) to distinguish between the 'real and the 'fake' day-0 data, while $\text{D}_2$ was fed with $\boldsymbol{Z}$ and

($G_2(\textbf{X})$) to distinguish between the 'real' and the 'fake' day-k data. Specifically, the discriminators would assign each sample a class label to tell if it belonged to the C-dimensional distribution of the real data ($\rho\left(\textbf{X}\right)$ or $\rho\left(\textbf{Z}\right)$) or from the distribution of the fake data generated by $G_1$ or $G_2$.

For the network training, we expected $G_1$ and $G_2$ to generate more convincing samples, while $D_1$ and $D_2$ to be more perceptive to better discriminate between the true and the fake samples. The performances of the networks in such contest could be quantified by adversarial losses. As with ADAN, here we adopted the mean absolute error (MAE), or $L_1$ loss, as the adversarial loss function. For $G_1$ and $D_1$, the adversarial loss can be expressed as follows:

$$\mathcal{L}_{adv}\left(D_1\right) = E_{\textbf{X} \sim p_{data}(\textbf{X})}\left[\|D_1\left(\textbf{X}\right) - b\|_1\right] + E_{\textbf{Z} \sim p_{data}(\textbf{Z})}\left[\|D_1\left(G_1\left(\textbf{Z}\right)\right) - a\|_1\right]$$
$$\mathcal{L}_{adv}\left(G_1\right) = E_{\textbf{Z} \sim p_{data}(\textbf{Z})}\left[\|D_1\left(G_1\left(\textbf{Z}\right)\right) - c\|_1\right] \tag{8}$$

where $a$ is the label for the fake neural firing rates, $b$ is the label for the real neural firing rates, and $c$ is the value that $G_1$ wants $D_1$ to believe for fake neural firing rates. Typically, we can set $a = 0$, and $b = c = 1$. For $D_2$ and $G_2$, the adversarial loss $\mathcal{L}_{adv}\left(D_2\right)$ and $\mathcal{L}_{adv}\left(G_2\right)$ have a similar form:

$$\mathcal{L}_{adv}\left(D_2\right) = E_{\textbf{Z} \sim p_{data}(\textbf{Z})}\left[\|D_2\left(\textbf{Z}\right) - b\|_1\right] + E_{\textbf{X} \sim p_{data}(\textbf{X})}\left[\|D_2\left(G_2\left(\textbf{X}\right)\right) - a\|_1\right]$$
$$\mathcal{L}_{adv}\left(G_2\right) = E_{\textbf{X} \sim p_{data}(\textbf{X})}\left[\|D_2\left(G_2\left(\textbf{X}\right)\right) - c\|_1\right] \tag{9}$$

The core idea of Cycle-GAN is to make the learned mapping functions cycle-consistent so as to reduce the space of possible mapping functions. As shown in (**Appendix 1—figure 1**), the two highlighted cycles should be able to bring the corresponding data back to the original domain, for example, the distribution of the recovered day-k neural firing rates $G_2(G_1(\textbf{Z}))$ should be similar to the distribution of the real day-k neural firing rates $\textbf{Z}$. Therefore, we define the cycle consistency loss as follows:

$$\mathcal{L}_{cyc}\left(G_1, G_2\right) = E_{\textbf{X} \sim p(\textbf{X})}\left[\|G_1\left(G_2\left(\textbf{X}\right)\right) - \textbf{X}\|_1\right] + E_{\textbf{Z} \sim p(\textbf{Z})}\left[\|G_2\left(G_1\left(\textbf{Z}\right)\right) - \textbf{Z}\|_1\right] \tag{10}$$

Note here we also applied the $L_1$ loss.

Taken together, the full loss function is written as:

$$\mathcal{L}\left(G_1, G_2, D_1, D_2\right) = \mathcal{L}_{adv}\left(D_1\right) + \mathcal{L}_{adv}\left(G_1\right) + \mathcal{L}_{adv}\left(D_2\right) + \mathcal{L}_{adv}\left(G_2\right) + \mathcal{L}_{cyc}\left(G_1, G_2\right) \tag{11}$$

and the training process is to solve this min-max optimization problem:

$$G_1^*, G_2^*, D_1^*, D_2^* = \arg \min_{G_1, G_2} \max_{D_1, D_2} \mathcal{L}\left(G_1, G_2, D_1, D_2\right) \tag{12}$$

**Appendix 1—table 2.** Cycle-GAN hyperparameters.

| parameter | value |
|---|---|
| Total number of trainable parameters | 74,208 |
| Batch size | 256 |
| Discriminator ($\textbf{D}_1$) learning rate | 0.01 |
| Discriminator ($\textbf{D}_2$) learning rate | 0.01 |
| Generator ($\textbf{G}_1$) learning rate | 0.001 |
| Generator ($\textbf{G}_2$) learning rate | 0.001 |
| Number of training epochs | 200 |

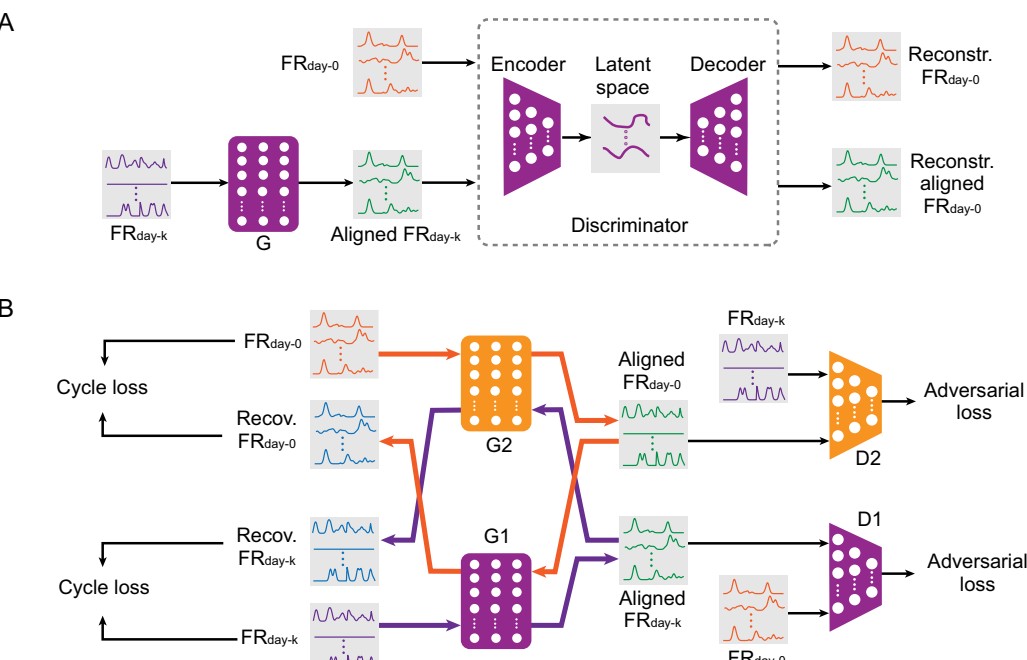

**Appendix 1—figure 1.** Adversarial neural networks proposed for iBCI stabilization. (**A**) The architecture of ADAN. A feedforward network (the generator, 'G') takes the neural firing rates on day-k ('FR$_{day-k}$') as input and applies a transform on them to produce the aligned neural firing rates ('Aligned FR$_{day-k}$'). Next, an autoencoder (the 'Discriminator') takes as input both the firing rates on day-0 ('FR$_{day-0}$') and the Aligned FR$_{day-k}$ and aims to discriminate between them, giving the adversarial loss. (**B**) The architecture of CycleGAN used as an aligner for an iBCI. A feedforward neural network ('G1') takes FR$_{day-k}$ as input and produces Aligned FR$_{day-k}$ after applying a transformation. Another feedforward network ('D1') aims to discriminate between Aligned FR$_{day-k}$ and FR$_{day-0}$; the performance of D1 contributes the first adversarial loss. A second pair of feedforward networks ('G2' and 'D2') function in the same way, but aim to convert FR$_{day-0}$ into an Aligned FR$_{day-0}$ that resembles FR$_{day-k}$; these contribute to the second adversarial loss. The discrepancy between the real FR$_{day-k}$ and Recovered FR$_{day-k}$ (generated by passing FR$_{day-k}$ through G1 followed by G2) contributes a cycle loss (and similarly for FR$_{day-0}$ and Recovered FR$_{day-0}$). The purple and orange arrows highlight these two cyclical paths through the two networks.

