## [Editor Report]

This paper reports a new way to deal with the drift of neural signals and representations over time in a BCI. Given the context of the rapidly advancing field, the reviewers assessed the findings to be useful and potentially valuable. With the code provided for other investigators to use, the strength of evidence was convincing.

---

## [Decision Letter]

**Decision letter after peer review:**

Thank you for submitting your article "Using adversarial networks to extend brain computer interface decoding accuracy over time" for consideration by *eLife*. Your article has been reviewed by 2 peer reviewers, including Caleb Kemere as the Reviewing Editor and Reviewer #1, and the evaluation has been overseen by Joshua Gold as the Senior Editor.

Essential revisions:

1) Recognizing that it may seem unfair given the length of time that your work has been in review, for the general *eLife* audience, the reviewers felt that it was required was to address the performance of the NoMAD approach (https://www.biorxiv.org/content/10.1101/2022.04.06.487388v1). Ideally, this would be a direct comparison. More generally, it would be valuable to discuss the relative merits of alignment approaches based only on the moment-by-moment cofiring of neurons (e.g., CycleGAN) versus alignment approaches which further leverage the dynamics in the latent space.

*Reviewer #2 (Recommendations for the authors):*

In this paper, Ma et al. tackle the problem of how to allow intracortical BCIs to sustain a high level of performance when there is changes in the neural signals recorded from the array and the behavior of the monkey. Such changes could be due to changes in signal quality, the tuning of the neurons, turnover of recorded neurons etc. In an ideal world, for patients using this day in and day out, there would be a quick approach to understand what the current state of the decoder is and quickly and readily adapt to the current setting so that the patient sees no drop in performance. This is a somewhat well studied question and barring older work, Stavisky, Sussillo et al. 2016 proposed a solution to this problem by using multiplicative recurrent neural networks (RNNs) that can select the best decoder given the neural data by learning from many different samples. The Miller lab in 2018 proposed using GANs to solve this problem, and again in collaboration with Dr. Pandarinath's lab has developed an approach using LFADS (called NOMAD, Karpowicz et al. 2022) to solve this problem. Here they use a different type of GAN to solve this problem. The paper is well structured, reasonably clear, the datasets are impressive and the authors have applied their approach to these datasets and compared to an approach which is based on factor analysis.

However, currently I am unsure the degree of advance provided by this paper. In particular, given that two of the datasets studied in this paper (Monkey J and Monkey C center out reach) are also present in the Karpowicz et al. 2022 paper, we need to rigorously compare both of them. The improvement from the ADAN approach seems somewhat minor in my opinion.

1. I find the results only modestly improve over their own existing approach (ADAN) and yes it does better than a simple factor analysis based method but that is simply stated as a powerful neural network is way better than a simple set of linear operations. I mean this is a little bit like a sprint race between me and Usain Bolt, there is just no contest there.

2. The related issue is that they are at best proposing a minor improvement over their own Cycle-GAN study. More worryingly, their approach does not seem to be better than the NoMAD Study from Karpowicz et al. 2022? I am all for many different approaches, but I am tad worried that there is just minimal improvement over and above their previous approach. It also feels like we are not performing a fair comparison to the state of the art, which some subset of authors in this paper has worked on! I think at a minimum they need to run NoMAD on the same datasets with whatever binsizes they choose and show that their method is comparable. I say this from the perspective that these are all offline decoding analyses and yes it is computationally expensive but does not need new experiments. In fact NoMAD runs better on this dataset with a 20 ms bin compared to a 50 ms bin.

Karpowicz et al. 2022 (bioRxiv), shares considerable author overlap with Ma et al. 2022

(Xuan Ma, Lee Miller)

The reference for this is totally mangled btw.

3. Why do I say this. Any reader who is aware of NoMAD would be like this is a strawman comparison. I think putting all of these methods on equal footing is necessary to move the field forward! I hope the authors don't feel like this is unreasonable. In addition it is the same data from I think a rockstar monkey J (95 days of data, similar task etc). Monkey J is also used in the NoMAD paper. So same dataset, multiple papers and two to three different methods :)! Figure 3 at a minimum needs a plot of the NoMAD results.

4. Of interest would be to discuss the number of parameters in each of these approaches. If the authors want, it might make sense to show how long it takes for PAF, ADAN, Cycle-GAN and NoMAD and this could be a supplementary figure. Maybe NoMAD will need way more training trials. It looks like PAF should have minimal parameters but Cycle-GAN is at least 2x as expressive as ADAN.

5. There is a theoretical point here. The GANs are trying to make the data indistinguishable from one another but as the neural data analysis shows the principal angle is still pretty substantial for 10 dimensions (~50 degrees). This will hurt their decoders. This might be an inherent disadvantage of GANs because they will likely stop once the data look like they are similar to the original distribution. But what you want is ideally something that adjusts the Day-k data to be near identical to the Day-0 data, in which case methods that maximize alignment might be a better approach. This should be discussed in the manuscript.

6. A weakness of all of these studies is that it is all done offline, what approach wins best online is an open question. Of note Stavisky, Sussillo et al. worked online. This should be a caveat in the discussion of these studies as it is an open question which of these approaches will be most successful online.

[Editors' note: further revisions were suggested prior to acceptance, as described below.]

Thank you for resubmitting your work entitled "Using adversarial networks to extend brain computer interface decoding accuracy over time" for further consideration by *eLife*. Your response and revised article has been evaluated by Joshua Gold (Senior Editor) and a Reviewing Editor as well as the original reviewers.

The reviewers appreciated your thorough responses to their comments. However, upon discussion, there was a consensus that two important issues remain that should be addressed:

1. The comparison to NoMAD seems important enough that adding to the manuscript details from the response letter (point #3 from R2) would be useful, particularly in terms of your contention that best within-time-bin alignment is likely a valuable component of more complex systems.

2. Given that this is a Tools and Resources article, we believe that the description of the approach in Appendix 4 is still insufficient. In addition, we request code or pseudo-code that implements those algorithms in a way that a community member would be able to rapidly use them.

---

## [Author Response]

Essential revisions:1) Recognizing that it may seem unfair given the length of time that your work has been in review, for the general eLife audience, the reviewers felt that it was required was to address the performance of the NoMAD approach (https://www.biorxiv.org/content/10.1101/2022.04.06.487388v1). Ideally, this would be a direct comparison. More generally, it would be valuable to discuss the relative merits of alignment approaches based only on the moment-by-moment cofiring of neurons (e.g., CycleGAN) versus alignment approaches which further leverage the dynamics in the latent space.

We appreciate the reviewers’ feedback, and we absolutely recognize the importance of being able to compare between different methods for neural representation alignment. While we would like to push back against the notion of NoMAD, or any one method, as being state-of-the-art for alignment, we have made an effort to respond to the reviewers’ concerns in our rebuttal to Reviewer #2 Points 2-3, and in the Discussion section of the manuscript.

Reviewer #2 (Recommendations for the authors):In this paper, Ma et al. tackle the problem of how to allow intracortical BCIs to sustain a high level of performance when there is changes in the neural signals recorded from the array and the behavior of the monkey. Such changes could be due to changes in signal quality, the tuning of the neurons, turnover of recorded neurons etc. In an ideal world, for patients using this day in and day out, there would be a quick approach to understand what the current state of the decoder is and quickly and readily adapt to the current setting so that the patient sees no drop in performance. This is a somewhat well studied question and barring older work, Stavisky, Sussillo et al. 2016 proposed a solution to this problem by using multiplicative recurrent neural networks (RNNs) that can select the best decoder given the neural data by learning from many different samples. The Miller lab in 2018 proposed using GANs to solve this problem, and again in collaboration with Dr. Pandarinath's lab has developed an approach using LFADS (called NOMAD, Karpowicz et al. 2022) to solve this problem. Here they use a different type of GAN to solve this problem. The paper is well structured, reasonably clear, the datasets are impressive and the authors have applied their approach to these datasets and compared to an approach which is based on factor analysis.However, currently I am unsure the degree of advance provided by this paper. In particular, given that two of the datasets studied in this paper (Monkey J and Monkey C center out reach) are also present in the Karpowicz et al. 2022 paper, we need to rigorously compare both of them. The improvement from the ADAN approach seems somewhat minor in my opinion.1. I find the results only modestly improve over their own existing approach (ADAN) and yes it does better than a simple factor analysis based method but that is simply stated as a powerful neural network is way better than a simple set of linear operations. I mean this is a little bit like a sprint race between me and Usain Bolt, there is just no contest there.

Although we acknowledge that nonlinear methods are theoretically superior to linear methods (at least when applied to nonlinear systems), we do not believe that this should detract from the significance of our paper. As the reviewer noted, the primary objective of our study was to compare a novel approach (Cycle-GAN) to two established techniques (ADAN and Procrustes alignment of factors), all intended to align neural data. While the best-case performance boost of Cycle-GAN over ADAN is not large (although see our second point below), here we note several reasons to believe that Cycle-GAN is a much more promising technique.

First, Cycle-GAN is much more robust to hyperparameter tuning than ADAN. This finding is not trivial, as GANs are notoriously difficult to train, as was the case for ADAN. Part of the reason we adopted Cycle-GAN was ADAN’s very poor performance (and our considerable concern) in our initial tests in applying it to a broader range of data. ADAN must be hand-tuned by someone with machine learning expertise for each new dataset. Our hyperparameter analysis (Figure 4) suggests that Cycle-GAN is more likely to be effective ‘out-of-the-box’, and at working with different data binning and sampling rates. In your sprint analogy, ADAN might be seen as a blindfolded Usain Bolt -- he might still outrun you, but only if his trainer leads him to the track and carefully lines him up facing in the right direction before the race.

Second, we have expanded our analysis in the revised manuscript to include neural alignment during continuous recordings, and demonstrated that Cycle-GAN performs much better than ADAN (and PAF) in that setting. In our original submission, we exclude periods of time when the monkey is not engaged in the task from the datasets used for alignment (for all three methods.) However, in a true iBCI setting, the user has uninterrupted control, so it would not be ideal to train the aligner excluding portions of a recording session that are not task related (as in our previous analysis). Our evaluation in the continuous recording setting is therefore a more accurate reflection of how each method might perform in a clinical setting- and here, the improvement of Cycle-GAN over ADAN is clear.

And third, as discussed in the manuscript and above, Cycle-GAN can be used directly with any previously trained spike-rate based decoder. This is in contrast to ADAN and PAF, which only align the neural latent space and therefore require either a new, latent space decoder to be trained, or an additional post-alignment, backwards-projection step to convert the latent representation into a predicted set of spikes. The backwards-projection step leads to lower decoding performance for ADAN, and complete failure for PAF, as shown in Appendix 3 – Figure 1. Thus, Cycle-GAN is a more versatile and practical solution because of its flexibility in integrating with other existing iBCI pipelines.

Taking all these points together, Cycle-GAN indeed represents a substantial improvement over existing techniques for improving the stability of iBCI decoders over time.

Finally, from a scientific perspective (and related to our third point), Cycle-GAN is interesting as it is the first neural alignment approach that has been demonstrated to perform well without relying on the computation (and stability) of latent manifolds. While not explored in this manuscript, this property might make Cycle-GAN of potential interest in applications where neural dynamics are higher-dimensional, such as in cognitive tasks (Rigotti et al., 2013).

Rigotti, M., Barak, O., Warden, M. R., Wang, X. J., Daw, N. D., Miller, E. K., and Fusi, S. (2013). The importance of mixed selectivity in complex cognitive tasks. Nature, 497(7451), 585-590.

Action in the text (pages 9-10): We have added some discussion points from the response above to the first Discussion section, as well as the ‘iBCI stabilization *without manifolds’* Discussion section, to better highlight the advantages of Cycle-GAN.

2. The related issue is that they are at best proposing a minor improvement over their own Cycle-GAN study. More worryingly, their approach does not seem to be better than the NoMAD Study from Karpowicz et al. 2022? I am all for many different approaches, but I am tad worried that there is just minimal improvement over and above their previous approach. It also feels like we are not performing a fair comparison to the state of the art, which some subset of authors in this paper has worked on! I think at a minimum they need to run NoMAD on the same datasets with whatever binsizes they choose and show that their method is comparable. I say this from the perspective that these are all offline decoding analyses and yes it is computationally expensive but does not need new experiments. In fact NoMAD runs better on this dataset with a 20 ms bin compared to a 50 ms bin.Karpowicz et al. 2022 (bioRxiv), shares considerable author overlap with Ma et al. 2022(Xuan Ma, Lee Miller)

(See next point for response)

The reference for this is totally mangled btw.3. Why do I say this. Any reader who is aware of NoMAD would be like this is a strawman comparison. I think putting all of these methods on equal footing is necessary to move the field forward! I hope the authors don't feel like this is unreasonable. In addition it is the same data from I think a rockstar monkey J (95 days of data, similar task etc). Monkey J is also used in the NoMAD paper. So same dataset, multiple papers and two to three different methods :)! Figure 3 at a minimum needs a plot of the NoMAD results.

We grouped the response to the reviewer’s two comments concerning NoMAD here, and have added text on this subject to the Introduction and Discussion of the text.

First, we would like to push back against the notion of NoMAD, or any other single study, as being state-of-the-art for alignment. An equal footing for these comparisons is indeed important, but there is as yet no consensus benchmark dataset or metric with which to contrast different alignment methods. This is why we make an effort in this paper to establish a rigorous framework to fairly compare multiple alignment methods, by (1) controlling for preprocessing and decoder architecture, (2) applying a fixed, appropriate set of evaluation metrics to a large ensemble of tasks and multiple monkeys, and (3) exploring other aspects of performance beyond accuracy, such as sensitivity to hyperparameters.

Comparison to NoMAD within this framework turns out to be problematic because NoMAD and Cycle-GAN are solving overlapping but different problems. A stable iBCI device has several interacting components: data preprocessing, an aligner that registers neural representations across days, and a decoder that translates neural activity to a predicted motor command. Higher iBCI performance could arise from an improvement to any of these processes. NoMAD includes the first two steps, performing both alignment of the neural representations via Kullback-Leibler Divergence (KLD) minimization and data preprocessing via LFADS-based smoothing. Because Karpowicz et al., contrast NoMAD (alignment + powerful dynamics-based smoothing) to two methods that perform alignment with only very simple linear smoothing, it is not possible to tell from their manuscript the extent to which NoMAD’s higher performance arises from better alignment vs their use of LFADS for data smoothing.

Nevertheless, the effects of the preprocessing can be inferred from their results: because of its more powerful dynamics preprocessing, NoMAD outperforms ADAN (and PAF) not only at day-k, but also on day-0, where no neural alignment is involved. The day 0 performance makes it clear that a substantial portion of NoMAD’s higher performance comes not from alignment but from how data are pre-processed.

We can also draw conclusions purely from the method NoMAD uses for alignment, namely by minimizing the KL divergence between the distributions of day-0 and day-k states that come out of a day-0 LFADS Generator network. This alignment strategy is very similar to the KLD minimization method tested in Farshchian et al., 2018; in that study, KLD minimization between neural latent states (obtained via an autoencoder) had inferior performance compared even to ADAN. This suggests that the apparent performance improvement of NoMAD over ADAN is a direct consequence of its embedded LFADS model rather than being due to a better alignment strategy. Theoretically, one could therefore replace the KLD-based alignment in NoMAD with a Cycle-GAN-based aligner and achieve even better performance.

It should also be noted that NoMAD cannot be used in combination with an existing neural decoder unless that decoder also gets (and was trained on) an LFADS-based smoothing.

Regarding authorship: as listed at the end of Karpowicz et al. 2022 (bioRxiv), Xuan Ma’s contribution was in data preparation, while Lee Miller’s contribution was in conceptualization, funding acquisition and manuscript revision. Neither author was substantially involved in the design and tests of the algorithms proposed in that work.

Action in the text (pages 9-10, lines 454-468): We have added some discussion points from the response above to the ‘Comparison of GANs to other methods for iBCI stabilization’ Discussion section.

4. Of interest would be to discuss the number of parameters in each of these approaches. If the authors want, it might make sense to show how long it takes for PAF, ADAN, Cycle-GAN and NoMAD and this could be a supplementary figure. Maybe NoMAD will need way more training trials. It looks like PAF should have minimal parameters but Cycle-GAN is at least 2x as expressive as ADAN.

Yes, PAF has a minimal number of parameters (102 ~ 103) as it is a classical linear algebra-based method. Cycle-GAN has roughly twice the parameters of ADAN due to the presence of an additional pair of generator and discriminator. For aligning two 96-channel neural datasets, Cycle-GAN has 74,208 parameters while ADAN has 35,946. We could not find information about the parameter count of NoMAD in the Karpowicz et al., preprint, however we expect it to be considerably higher, as it encompasses LFADS-based smoothing in addition to alignment.

We already provide a comparison of PAF, ADAN and Cycle-GAN training times in Figure 4B of our manuscript. We note that while Cycle-GAN training time for a given batch size is longer, is can actually be faster to train in practice because Cycle-GAN can be trained using larger batch sizes, whereas ADAN must be trained with small batch sizes to properly converge (as shown in Figure 4A). Again, we could not find any information about training time in the pre-print of NoMAD.

Finally, as stated earlier, we note that while forward-backward mapping between days does occur during Cycle-GAN training, only the forward mapping is performed during inference. Because of this, the inference speed for Cycle-GAN is comparable to that of ADAN, with both models completing the forward map of one sample (one 50 ms-binned vector of firing rates) in well under 1 ms.

Action in the text (Appendix 4 Tables 1-2): We have added a summary table including the total number of parameters for each method (and the related hyperparameter values) in Appendix 4 (ADAN: Appendix 4 Table 1; Cycle-GAN: Appendix 4 Table 2).

Action in the text (page 7, lines 303-306): added inference time for Cycle-GAN and ADAN.

5. There is a theoretical point here. The GANs are trying to make the data indistinguishable from one another but as the neural data analysis shows the principal angle is still pretty substantial for 10 dimensions (~50 degrees). This will hurt their decoders. This might be an inherent disadvantage of GANs because they will likely stop once the data look like they are similar to the original distribution. But what you want is ideally something that adjusts the Day-k data to be near identical to the Day-0 data, in which case methods that maximize alignment might be a better approach. This should be discussed in the manuscript.

It is important to note here that ~50 degrees is not THE principal angle between two 10-dimensional spaces: it is the 10th-smallest principal angle between those spaces. For any pair of N-dimensional hyperplanes, there are N principal angles that exist between them. Following the approach outlined in Knyazev and Argentati, 2002 and Elsayed et al., 2016, these are computed as follows: first, we reduce each sample (here the Day-X neuron firing rates) to 10 dimensions using PCA. Next, recursively for each n = 1…10, we identify the pair of principal vectors that are separated by the smallest angle and perpendicular to the prior selected pairs, and report that angle. When two hyperplanes are well-aligned, the leading principal angles between them can be very small, but often the last few angles are quite large.

Figure 7B shows that the ten principal angles between sessions many days apart (gray curves) are all substantially larger than the correspondingly ordered principal angles obtained between the day-0 neural recordings of even-numbered trials and odd-numbered trials (this was done to reduce the effect of any within-day drift). After processing by ADAN and Cycle-GAN, the principal angles become considerably smaller. For ADAN (red curves), the first three principal angles are close to the within-day values, but the remaining angles are much larger. In contrast, for Cycle-GAN, all principal angles except the last are even smaller than the within-day principal angles! This suggests that the aligner has effectively maximized the alignment of day-0 and day-k data, indeed making them as well or better aligned then two samples from the same day.

Action in the text (page 15, lines 727-752): We clarified our description of the principal angles computation and its interpretation in the related Methods section.

6. A weakness of all of these studies is that it is all done offline, what approach wins best online is an open question. Of note Stavisky, Sussillo et al. worked online. This should be a caveat in the discussion of these studies as it is an open question which of these approaches will be most successful online.

We believe that offline analyses are actually essential for the development of BCI systems because they allow for a comprehensive evaluation of the efficacy of a BCI algorithm without involving the complication of a user who can learn and adapt. We intend to add online tests in future work, and have allocated resources to develop online BCI platforms for both monkey and human subjects, This, however, is well out of the scope of what we can report in the current manuscript.

As discussed in the ‘Comparison of GANs to other methods for iBCI stabilization’ section, the approach taken by Stavisky, Sussillo et al., differed fundamentally from all our tested methods. They harnessed a large training dataset recorded over a span of many months to build a robust decoder for the monkey. While their results were impressive, it would be challenging to accumulate such a vast amount of training data in many applications, including those with paralyzed humans in the loop.

[Editors' note: further revisions were suggested prior to acceptance, as described below.]

The reviewers appreciated your thorough responses to their comments. However, upon discussion, there was a consensus that two important issues remain that should be addressed:1. The comparison to NoMAD seems important enough that adding to the manuscript details from the response letter (point #3 from R2) would be useful, particularly in terms of your contention that best within-time-bin alignment is likely a valuable component of more complex systems.

Comparison to NoMAD: as requested, we have added text from our previous response letter (Reviewer 2 point #3) to the Discussion of the manuscript. This can be found on lines 456 – 483.

2. Given that this is a Tools and Resources article, we believe that the description of the approach in Appendix 4 is still insufficient. In addition, we request code or pseudo-code that implements those algorithms in a way that a community member would be able to rapidly use them.

Formatting of appendices: we have converted all appendix figures into supplements of main figures in the text, and have integrated the text of appendices 1 and 2 into the figure legends and methods section. As discussed in my call with Josh Gold on June 1, Appendix 4 is far too bulky to include in the main text, so we have kept its appendix status here (submitted as “Supplementary File – Appendix 1”). We have also extended this appendix to include a more “practical” walk-through of how Cycle-GAN works. This section takes the reader through the steps of our Jupyter notebook tutorial posted at https://github.com/limblab/adversarial_BCI/, summarizing what is done in each code block of that notebook. With the provided tutorial notebooks, their summary in the Appendix, and the additional theoretical documentation in the Appendix, we believe that our manuscript now provides readers with ample support to apply Cycle-GAN alignment to their own data.